# Examining the relationship between language development, executive function, and screen time: A systematic review

**Mazhar Bal**[1], **Ayşe Gül Kara Aydemir**[2]*, **Gülüzar Şule Tepetaş Cengiz**[3], **Ahmet Altındağ**[4]

1 Department of Turkish and Social Sciences Education, Faculty of Education, Akdeniz University, Antalya, Turkey, 2 Department of Educational Sciences, Akdeniz University, Antalya, Turkey, 3 Child Development Department of Bolu Abant İzzet Baysal University, Mehmet Tanrıkulu Vocational School of Health Services, Bolu, Turkey, 4 Faculty of Education, Department of Educational Sciences, Bolu Abant İzzet Baysal University, Bolu, Turkey

* agkara@gmail.com

**Data Availability Statement:** https://osf.io/n7uc6/?view_only=8987e34f20da40ddba6c6c7c7009e84d

**Funding:** The author(s) received no specific funding for this work.

## Abstract

This systematic review study examines the relationship between language development, executive function, and screen time in early childhood. The early childhood period is a crucial phase for the development of the brain, during which fundamental language and executive function skills undergo rapid evolution. This review synthesizes findings from 14 peer-reviewed studies that focused on language development, executive function, and screen time together to provide a comprehensive understanding of their relationship. The findings of current study were categorized under four themes: screen content and adherence to guidelines, parent-child interaction and the family context, passive and active screen time, and attention issues. The findings suggest that interactive and educational screen content may positively influence language development and executive functions when aligned with recommended screen time guidelines. In contrast, excessive passive screen time, such as watching television, has been associated with negative impacts on cognitive and social skills, particularly affecting attention, memory, and emotional regulation. The prevalence of attention problems is found to be higher in individuals who engage in high levels of screen time. This highlights the necessity for balanced consumption of screen media. The review emphasizes the pivotal role of parent-child interaction, where high-quality engagement and verbal scaffolding during screen time can mitigate adverse effects. Furthermore, socioeconomic and cultural factors also play a significant role. Higher socioeconomic status (SES) families are better able to manage screen time and leverage educational content to support development. These findings have the potential to inform the actions of parents, educators, and policymakers. Adherence to recommended screen time guidelines can mitigate the potential negative impact on executive functions and language skills. Furthermore, the importance of limiting passive screen time and ensuring a balance between screen use and real-world interactions and play opportunities is also highlighted.

**Competing interests:** The authors have declared that no competing interests exist.

## Introduction

The early childhood period (0–78 months), extending from birth to preschool years, represents a critical phase in language development. During this age range, children rapidly acquire fundamental language skills, beginning with comprehension and expression capabilities, gradually progressing to sentence construction and understanding structural rules of language [1, 2]. This developmental process significantly influences not only linguistic abilities but also social, emotional, and cognitive development [3, 4].

Examining the theoretical foundations of language development, Piaget's cognitive development theory and Vygotsky's sociocultural approach are closely associated with the developmental characteristics of children aged 0–78 months. While Piaget emphasized the critical role of language skills in developing thinking, problem-solving, and analytical abilities during this age range [5, 6], Vygotsky highlighted the importance of language in social interaction, emphasizing its role in developing communication and emotional bonds [7, 8]. Furthermore, language functions as a crucial tool in children's cultural identity formation and environmental interpretation during this early period [9, 10].

Contemporary neurobiological research reveals a strong connection between language processing and executive functions (EF) during early childhood (0–78 months) [11, 12]. Language development plays a crucial role in shaping children's cognitive processes and expressing thoughts [13, 14]. Executive functions are critical in developing language abilities and interpreting complex linguistic structures [15–20]. During this critical developmental period, executive functions encompass higher-order cognitive abilities such as planning, goal setting, problem-solving, and memory, essential for regulating thought processes [21, 22]. Studies indicate that executive functions and language development mutually reinforce each other during this age range [23, 24].

Screen time can both positively and negatively impact language and cognitive development in children aged 0–78 months [25, 26]. While excessive screen time may delay language acquisition [27], moderate screen use with educational content and parental guidance can support language development and certain executive functions [28, 29]. There are varying interpretations regarding the definition and effects of screen time [30]. Given these mixed results, understanding this relationship is crucial for maximizing benefits while minimizing risks [31–33]. According to the World Health Organization, screen time refers to passive exposure to screen-based entertainment [34]. It is commonly defined as time spent accessing media through various devices, including television, mobile phones, DVDs, computers, and video games [35]. Research indicates that screen media has become a regular part of daily routines, particularly for children under 36 months [36, 37], with screen use frequency increasing for children aged 18–36 months [38]. Findings regarding screen use effects in this age group are contradictory. While some studies demonstrate potential benefits from educational content and moderate use [39–42], others reveal negative impacts on attention, working memory, and problem-solving skills, particularly with prolonged use [43–46]. Bukhalenkova and Almazova [46] conducted a study involving 772 mothers of children aged 5–6 and examined the relationship between active screen time factors such as time spent playing computer games and parental involvement in children's computer games, concluded that long-term screen use limits children's creativity and problem-solving skills. Involving 106 parents (aged 18 years or older) of children between 0 and 4 years residing in Australia, Halpin and their colleagues [47] argued that long-term screen use weakens children's time management skills. Furthermore, prolonged screen exposure has been demonstrated to negatively affect children's ability to focus and plan tasks [48], as well as their capacity to recognize and regulate their emotions [49]. McArthur, Tough & Madigan [50] involved 1994 mothers and their children in Calgary, Canada, they

evaluated children's screen time, behavior problems, developmental milestones, and vocabulary acquisition at 36 months based on maternal reports.

Systematic reviews and meta-analyses have extensively examined the impact of screen time on child development in the 0–78-month age range. While Jing et al. [51] focused on screen media exposure and vocabulary development, McHarg et al. [52] and Likhitweerawong et al. [29] investigated the relationship between screen time and executive functions. Bustamante et al. [53] conducted a meta-analysis examining the impact of screen time on executive functions in children under 6 years, revealing the complex nature of this relationship. Li et al. [54] provided a broader perspective through their meta-analysis of early childhood screen use and health indicators. Shokrkon and Nicoladis [17], in their literature review examining the directionality between executive functions and language skills, emphasized the complexity of this bidirectional relationship, though they did not address the role of screen time in this interaction. Building on these foundations, several recent studies have further investigated specific aspects of screen time's impact. Ponti [55], in a comprehensive review for the Canadian Pediatric Society, examined how digital media affects child development, emphasizing the need for age-appropriate content and guided interaction. Jannesar et al. [56], in their scoping review of screen time's effects on children's brain development, highlighted the importance of neurological impacts but did not specifically examine the interaction between language development and executive functions. Panjeti-Madan and Ranganathan [57] provided a detailed analysis of screen time's impact on multiple developmental domains, including cognitive, language, physical, and social-emotional areas, but did not specifically examine the interrelationship between these domains. Regarding language development specifically, Bhutani et al. [58] conducted a scoping study examining how screen time affects children's language development. Oktarina et al. [59] systematically reviewed the relationship between screen time and children's language development, demonstrating how this relationship varies by age, content, and parental involvement. Their findings indicate a complex relationship mediated by multiple factors, though they did not investigate the role of executive functions in this relationship. Gowenlock et al. [60] further contributed to this understanding through their comprehensive review of video exposure's impact on language development in children aged 3–11 years, emphasizing the importance of content quality and interaction patterns. Additionally, Aziz et al. [61] provided valuable insights in their study examining screen exposure's impact on language development among toddlers and preschoolers, emphasizing the need for controlled screen time and quality content.

Understanding the complex interaction between screen time, language development, and executive functions is critically important, particularly for children aged 0–78 months. While existing literature has typically examined these three factors in the context of binary relationships, their trilateral interaction remains insufficiently addressed, representing a significant research gap. This systematic review aims to fill this gap by synthesizing current evidence regarding the relationship between screen time and language development and executive functions in early childhood (0–78 months). The study's findings will provide evidence-based guidance to parents and educators regarding optimal screen time use while contributing to the development of scientifically based guidelines for technology integration in preschool educational institutions. Another significant aspect of the study is providing evidence-based data needed for developing educational policies regarding technology use in early childhood. This data will enable the development of strategies that maximize positive effects while minimizing negative impacts of screen use.

## Purpose of the study and the research questions

Although technology has many benefits and risks that can negatively affect children, completely excluding today's children from the use of technology is like neglecting to teach an

island child how to swim. It is of critical importance to understand the complex relationship between screen time, language development, and executive functions in early childhood in the context of the digital age. Given the virtual possibility of children interacting with technology, it is imperative for parents, caregivers, and teachers to be aware of potential consequences of technology on children's development, and to act responsibly. Mantilla and Edwards [62] noted that digital technology has now become and is accepted as an integral part of young children's daily life; setting from this point the way it is utilized by and with young children play a pivotal role in their well-being, safety, communication, and learning. This study is significant in terms of addressing the gap between cognitive psychology and the impact of technology on child development. The early childhood period is a critical period for brain development, during which basic language and executive function skills rapidly develop. Excessive screen time, which often replaces interactive and stimulating activities, has been shown to negatively affect these developmental processes [36, 53, 63, 64]. In contrast, purposeful and controlled screen time could support cognitive growth if appropriately integrated into educational frameworks [65, 66]. Given the pervasiveness of digital technology in the lives of young children, this research attempts to address the relationship between language development, executive function, and screen time by synthesizing existing findings from the literature to provide a comprehensive understanding of how these interactions influence developmental outcomes. The objective of this study is to present implications for parents, educators, and policy makers regarding the potential risks and benefits associated with the use of digital media. The findings can inform the development of balanced media consumption guidelines and interventions designed to support healthy development. The findings of this study are expected to contribute to the formulation of strategies that leverage the educational potential of technology while mitigating its adverse effects on the cognitive and language development of young children. The study's focused examination of the relationship between language development, executive functions, and screen time renders it both timely and important. Consequently, the interplay among these three elements has been subjected to a comprehensive analysis. Accordingly, the overarching research question guiding the current study is formulated as below:

What is the relationship between screen time, language development, and executive functions in children aged 0–78 months as reported in the present research literature?

## Materials and methods

This study applied the principles of the PRISMA statement [67]. Additionally, the researchers used the CONSORT checklist adapted by Angosto, García-Fernández, Valantine, and Grimaldi-Puyana [68] to assess the reporting quality of the selected studies. The dataset used in this study is provided as **S1 File**. It is an Excel file containing all the necessary data to replicate the analyses and findings presented in this manuscript. The PRISMA checklist used in this study is provided as **S2 File** and outlines compliance with PRISMA guidelines. Both files have been deposited in the Open Science Framework (OSF) as supporting information. They can be accessed via the following link: https://osf.io/n7uc6/?view_only=8987e34f20da40ddba6c6c7c7009e84d.

### Search strategy

There were no restrictions on the search timeframe. A comprehensive search was conducted until October 1, 2023, for studies completed up to that date. To locate articles for systematic review, the researchers concentrated on studies published in English language, in journals indexed in a variety of databases, including Web of Science, Scopus, Education Resources Information Center (ERIC), EBSCO, and PubMed. In the preliminary search, experts employed

specific keywords. The keywords used in the systematic search included "Language Development," "Language Skill," "Language Teaching," "Language Education," "Executive Function," "Screen Time," "Children," "Pre-school Children," and "Early Childhood." These terms were combined using "AND" and "OR" conjunctions, and the population of interest was limited to preschool-aged children. The term "screen time" is used since it encapsulates a broad spectrum of devices (e.g., computers, televisions, phones) and their diverse uses (e.g., gaming, social communication) [69]. Our search strategy was designed to capture studies that fall within this expansive scope, assessing the impact of "screen time" on children's language development and executive functions. We used the term "screen time" to focus on the duration of engagement with electronic devices. This term is expected to encompass much of the extensive literature in this domain due to its representation of the time spent using electronic devices.

## Eligibility criteria

This systematic review examines all peer-reviewed empirical studies that specifically investigate the relationship between executive function, language development, and screen time during early childhood. Inclusion and exclusion criteria were established for the selection of studies to ensure that the identified articles were relevant to the purpose of the study. Duplicate studies were excluded. Table 1 presents the criteria in detail.

The article selection process took place in two stages. The first stage involved screening studies based on titles, abstracts, and full texts. Two experts (MB and AGKA) independently conducted the selection and screening process, being unaware of each other's decisions. The second stage involved a thorough review of the selected articles to ensure compliance with the inclusion criteria. Disagreements between the two reviewers were resolved by a third expert (GŞTC) acting as a reviewer. Studies that did not address the relationship between language development, executive function and screen time in early childhood were excluded because they did not meet the eligibility criteria.

## Search and selection process

For this systematic analysis, a total of 310 studies were initially identified. Of these, only 191 English articles were included in the study. Among these, 83 duplicate articles were identified and excluded. After reviewing the titles, 54 articles were excluded as they were irrelevant. Similarly, abstracts were analyzed, and 27 articles were excluded. The full texts of the remaining 23

**Table 1. Inclusion and exclusion criteria.**

| Category | Inclusion criteria | Exclusion criteria |
|---|---|---|
| Participants | Studies included children ranging in age from 0 to 78 months. | The studies included participants who were not in the early childhood age range. |
| Context | Studies focused on research on early childhood. Studies that examine the relationship between language development, screen time, and executive function. | Studies not addressing the relationship between language development, screen time and executive function in early childhood. |
| Study design | Original articles, qualitative studies, quantitative studies, mixed methods studies. | Book chapters, systematic reviews, scoping reviews, meta-analysis. |
| Language | English | Languages other than English |
| Keyword combinations | "Language Development" OR "Language Skill" OR "Language Teaching" OR "Language Education" AND "Screen Time" AND "Executive Function" "Children" OR "Pre-school Children" OR "Early Childhood" | Studies not containing the specified keyword combinations. |

articles were analyzed. Of the 9 studies, 5 were excluded as reviews and 2 were excluded because they did not meet the relevant criteria. After applying the exclusion criteria, a total of 14 articles [19, 66, 70–81] met all the specified criteria.

To ensure transparency and reproducibility, a detailed table documenting the evaluation process has been prepared and uploaded to the OSF for public access as supporting information. The table, entitled "**S1 Table**" presents a complete list of the studies reviewed during the screening stages, their classification as either "included" or "excluded," and providing rationale for their exclusion at various stages of the evaluation process. Furthermore, the table provides essential details such as the titles of the studies, digital object identifiers (DOIs) or uniform resource locators (URLs), the stages of the evaluation process, and the final inclusion or exclusion status of the studies. The aforementioned table can be accessed via the OSF at the following link: https://osf.io/n7uc6/?view_only=8987e34f20da40ddba6c6c7c7009e84d.

### Data extraction processes for analysis

In the final review, 14 articles were included, and a table was created to facilitate understanding of the information for the articles (see Table 2). The data presented in the table details the following: (a) authors; (b) year; (c) main focus of the study; (d) research questions; (e) measured variables, (f) methodology, and (g) participant characteristics covered in the studies.

To further ensure transparency and reproducibility, an additional table titled "**S2 Table**" has been prepared and uploaded to the OSF as supporting information. Table 2 provides detailed insights into the studies included in the review, whereas S2 Table is concerned with documenting the process of data extraction. The table includes the following: (a) the names of the data extractors; (b) the dates of data extraction; (c) confirmation of eligibility for inclusion; and (d) the extracted data variables necessary to replicate the analyses. This S2 Table is accessible on OSF at https://osf.io/n7uc6/?view_only=8987e34f20da40ddba6c6c7c7009e84d.

Additionally, the methodological quality and evidence strength of the included studies were evaluated using the GRADE (Grading of Recommendations Assessment, Development, and Evaluation) system, which assesses study limitations, inconsistency, indirectness, imprecision, and publication bias. Detailed GRADE ratings are provided in "**S3 Table**" as supporting information. S3 Table is available on the OSF at https://osf.io/n7uc6/?view_only=8987e34f20da40ddba6c6c7c7009e84d.

### Review of the quality of reporting

The articles included in this study were all quantitative in nature. The selected studies were initially reviewed using an adapted version of the Consolidated Standards of Reporting Trials (CONSORT) checklist by Angosto, García-Fernández, Valantine, & Grimaldi-Puyana [68], originally designed by Schultz et al. [82]. The original CONSORT statement [82] consists of 25 items and provides a general standard for reporting randomized controlled trials (RCTs). The adapted version incorporates methodological enhancements and expansions. Furthermore, the range and reliability of the data collection tools have been subjected to additional scrutiny. It provides a more suitable framework for evaluating different data collection tools and enhancing the reliability of the results. Additionally, the adapted version has been tailored to specific contexts and topics, making it more appropriate for the objectives of this study. The study aims to analyze the complex relationship between screen time, language development, and executive functions. The adapted version has allowed for a review of the reporting quality in the analyzed studies, with the goal of increasing the validity and reliability of the study findings. The tool includes a total of 20 items in the categories of "title and summary", "introduction", "methods", "results", "discussion", and "other information". The interrater agreement for

**Table 2. Descriptive register of articles.**

| Authors | Year | Main Focus | Research Questions | Measured Variables | Methodology | Participant Characteristics |
|---|---|---|---|---|---|---|
| Zhang et al. [66] | 2022 | Screen time, language development, executive function | Examine the associations between different types of screen time and indicators of cognitive development of preschoolers in Canada | Screen time, adherence, vocabulary, memory, age, sex, ethnicity, parental education, childcare group | A 6-month quasi-experimental pre–post-design study | 97 preschoolers aged 36–60 months from Alberta and Ontario, Canada. |
| Hutton et al. [75] | 2020 | Screen time, language development, executive function | What are the associations between screen-based media use and the integrity of brain white matter tracts supporting language and literacy skills in preschool-aged children? | Measured variables include ScreenQ, CTOPP-2, EVT-2, GRTR scores, FA, and RD in white matter tracts. | A cross-sectional study of healthy children conducted from August 2017 to November 2018. | 47 healthy children aged 3–5 years (57% girls), mean age 54.3 months (SD 7.5), recruited from a US children's hospital and primary care clinics. |
| Oflu et al. [78] | 2021 | Screen time, language development, executive function | Is excessive screen time associated with emotional lability in preschool children aged 2 to 5 years? | Screen time (low vs. high), sociodemographic (age, gender, parental age, education, mother's work, family structure, number of children, family type, caregiver, income), screen exposure (age at first exposure, co-viewing, reaction to limits, delaying needs), and emotion regulation (ERC: lability/ negativity, regulation). | Cross-sectional study (Jan 1—Mar 1, 2018) at a university hospital. Excluded: prematurity, low birth weight, no breastfeeding, language delays, chronic illness. | 240 healthy children aged 2–5 with screen time <1 hour or >4 hours. Excluded: prematurity, low birth weight, no breastfeeding, language delay, chronic illness. Sociodemographic: age, gender, parental age, education, mother's work, family structure, number of children, family type, caregiver, income. |
| Zhang et al. [81] | 2022 | Screen time, language development, executive function | What are the longitudinal associations of subjectively measured physical activity and screen time with multiple domains of cognitive development in young children? | Physical activity (organized, non-organized), screen time (TV, video games), total physical activity and screen time, screen time adherence, working memory, inhibitory control, intellectual ability (language, reasoning, speed). | Cross-sectional design was employed. | 96 children aged 2.5–5 years, English-speaking parents, excluding conditions limiting physical activity or cognitive performance, conducted in Edmonton, Canada. |
| Hendry et al. [73] | 2022 | Screen time, language development, executive function | How does the variation in the home environment, influenced by factors such as socioeconomic status, parental attitudes, and activities during the 2020 COVID-19 pandemic, relate to infants' emerging executive functions? | Parental attitudes towards learning, affection, attachment, engagement in activities, and screen use. | Recruited participants online, used linear regression and path analyses to study SES, parental attitudes, activities, and infant EFs during COVID-19. | 575 UK infants aged 8–36 months, from diverse SES. |
| Dolgikh et al. [72] | 2023 | Screen time, language development, executive function | What is the impact of attending extra classes on the development of executive functions in preschool children? | NEPSY-II subtests: Inhibition, Statue, Memory for Designs, Sentences Repetition, and Dimensional Change Card Sort to measure executive functions in preschoolers. | Compared executive functions in children attending vs. not attending extra classes using subtests and Mann-Whitney U-test. | 124 Russian children (mean age 78 months), 17% boys, 80% medium-income, 1,000– 1,300 minutes weekly screen time, differing maternal education. |
| Hu et al. [74] | 2020 | Screen time, language development, executive function | (1) Is there a difference in Chinese children's screen time in terms of their gender (male or female), community (urban or rural), and sibling status (one child or multiple children)? (2) Is there a relationship between Chinese children's social and cognitive development and their active or passive screen time? | Receptive vocabulary, math achievement, science knowledge, executive function, and social skills. | Quantitative study using stratified random sampling and hierarchical multiple regression. | 579 five-year-old children from rural Guangdong, China, including boys and single-child households, with potentially high screen time. |

(*Continued*)

**Table 2.** (*Continued*)

| Authors | Year | Main Focus | Research Questions | Measured Variables | Methodology | Participant Characteristics |
|---|---|---|---|---|---|---|
| Veraksa et al. [19] | 2021 | Screen time, language development, executive function | (1) Is there a difference between developmental outcome for active and passive screen time in preschool children? (2) Is watching television (passive screen time) a negative predictor for the development of phonological memory in children? (3) Is the interaction of children with smart electronic devices an auxiliary factor for the development of phonological memory in children? (4) Is there an influence of family factors on the development of phonological memory in preschoolers? | Daily passive and active screen time, phonological memory, age, Raven CPM score, mother's education, family income. | The study was conducted in two stages: at Time 1, the association between children's phonological memory, passive and active screen time, and family factors was examined. At Time 2, one year later, the impact of passive and active screen time on individual progress in phonological memory development was evaluated. The following test were administered: Understanding of Similar Sounding Words (USSW) Raven's Colored Progressive Matrices (CMP) | 122 preschool children (mean age 5.72, SD 0.33), 54.9% boys. Mothers' education: 79.51% higher, 2.46% academic, 4.92% incomplete higher, 11.48% secondary specialized, 1.63% secondary general. Family income: 4.96% insufficient, 13.22% low, 78.51% average, 3.31% high. |
| Cliff et al. [71] | 2017 | Screen time, language development, executive function | What are the associations between physical activity, screen-based entertainment, and various developmental and health outcomes in preschool-aged children? | Physical activity, screen time, executive function, language, self-regulation, social and emotional development, empathy, cardiac outflow, anthropometrics, retinal microvasculature, blood pressure. | Prospective cohort study of 430 children (ages 3–5) assessing physical activity, cognitive and language development, psychosocial development, adiposity, and cardiovascular health, with input from educators and parents. | 430 healthy, typically developing children (ages 3–5) in ECEC centers. Data collected on caregiver demographics, household characteristics, media device ownership, and parental health history. |
| Kim & Chung [76] | 2021 | Screen time, language development, executive function | What is the association between exposure to television or video (TV time) and children's language development and school achievement in childhood? | TV time, parental education, household income, early development (Denver II, K-ASQ), language development (REVT), school achievement (parent/teacher evaluation). | Annual assessments from birth to 87.9 months using parental questionnaires for TV time, group-based trajectory analysis, and various tools for language development and school achievement. Data from the Panel Study on Korean Children, following ethical standards. | 1087 participants (birth to 87.9 months) sampled via stratified multi-stage sampling. TV time reported from ages 2.2 to 7.3, categorized into four groups with differing socioeconomic variables. **Note:** *This article is included to the study since it shows the progression from early childhood (starting at 5.5 months) to later stages (up to 87.9 months) allows for a comprehensive understanding of how screen exposure impacts cognitive and language development over time.* |
| Medawar, et al. [77] | 2023 | Screen time, language development, executive function | How do mothers' home literacy beliefs and practices, along with screen media exposure, influence language outcomes in Argentinean toddlers? | Measured variables: exposure time to devices (TV, cell phone, computer, tablet), activity types, content exposure frequency, and shared device and reading material use. | Selected 465 mothers via social networks, used hierarchical linear regressions, checked model assumptions, added child demographics, HLE variables, and screen media measures, examined mediation and moderation effects. | 465 Argentinean mothers of Spanish-speaking children (18–36 months, 49.2% girls, mean age 26.35 months, SD 5.04). Most parents had tertiary or university education (mothers 93.54%, fathers 80%). |

(*Continued*)

**Table 2.** (Continued)

| Authors | Year | Main Focus | Research Questions | Measured Variables | Methodology | Participant Characteristics |
|---------|------|-----------|-------------------|-------------------|-------------|----------------------------|
| Supanitayanon, et al. [80] | 2020 | Screen time, language development, executive function | What are the associations between screen media exposure variables in the first 2 years of life and cognitive development at 4 years of age, considering positive parenting behaviors? | Measured variables: age of screen media onset, cumulative screen exposure, caregiver-child verbal interaction during screen time, and types of devices used (TV, handheld, computers). | A longitudinal cohort study to investigate the relationship between screen media exposure, parenting behaviors, and cognitive development in young children. | 274 healthy infants with typical development, full-term births ($\geq$2500 g), no complications or medical illnesses, normal growth verified by CAT/CLAMS, middle to high SES in Thai context. |
| Carson & Kuzik [70] | 2021 | Screen time, language development, executive function | What is the association between parent-child technology interference and cognitive and social-emotional development in preschool-aged children? | Parent-child technology interference (six devices), cognitive development (memory, inhibition, vocabulary), social-emotional development (sociability, externalizing/internalizing behaviors, prosocial behavior, self-regulation). | Assessing parent-child tech interference, measuring cognitive development with iPad tasks, evaluating social-emotional development via questionnaire, and using multiple linear regression analyses with adjustments. | 100 preschool children (3–5 years) and parents from Edmonton, Canada. Average child age: 4.5 years (29% female). Average parent age: 37.5 years (80% mothers). |
| Ribner et al. [79] | 2021 | Screen time, language development, executive function | Investigate whether the content and context of media use—educational, entertainment, and background television—is related to children's language and literacy skills. | Foreground TV (educational and entertainment), background TV, self-regulation issues, language skills, literacy skills. | Path models were used to estimate the direct and indirect associations between context and content of media use with language and literacy skills. | English-speaking families with 922 children (8 months to 8 years), including 461 girls, mean age 66.19 months (3–7 years). |

Source: Own elaboration.

the CONSORT protocol ratings in our systematic review study was high across the various items assessed. Each study was independently scored by two researchers who evaluated the sections they reviewed and rated each item with a score of 1, indicating whether the study satisfied the criterion or whether the item was applicable to the study. There was disagreement between the researchers in the evaluation of three studies. Disagreements were resolved by checking and discussing the original study until consensus was reached between the researchers. The first researcher is an expert in the field of language development in the preschool period, and the second researcher is an experienced researcher in the field of instructional technology.

The agreement rates, as reported in the last row of Table 3, indicate a range from 85% to 100% among the different studies evaluated. Specifically, seven studies achieved an agreement rate of 90%, three studies achieved a rate of 95%, two studies reached 85%, and two studies attained a perfect agreement rate of 100%. This high level of agreement demonstrates the consistency and reliability of the ratings provided by the reviewers.

To check the quality of reporting of the studies, the quality of reporting of the 14 quantitative studies evaluated in the research was evaluated out of 20 points, presented in Table 3:

As shown in Table 3, 1 article received 20 points [71], 2 articles received 19 points [72, 80], 2 articles received 18 points [66, 77], 4 articles received 17 points [19, 74–76], 4 articles received 16 points [70, 73, 79, 81], 1 article received 15 points [78]. Based on these data, all the articles have a value equal to or greater than 15 points. The average score of the articles was 17.21. In the majority of cases, the studies were in accordance with the standards for acceptable reporting, in terms of the design, methodology and implementation. Data collection tools also provide information about the quality of reporting of the studies. The studies employed varying numbers of data collection tools as follows: 1 [73, 77], 2 [19, 66, 72, 78, 81], 3 [70, 75, 76,

**Table 3. Reporting quality of the articles.**

| No Ítem | SECTION, TOPIC, ITEM LIST | Zhang etal. (2022a) | Hutton etal. (2020) | Ribner etal. (2021) | Ofiu etal. (2021) | Zhang etal. (2022b) | Hendry etal. (2022) | Dolgikh etal. (2023) | Hu etal. (2020) | Veraksa etal. (2021) | Cliff etal. (2017) | Kim & Chung (2021) | Medawar etal. (2023) | Supanitayanon etal. (2020) | Carson & Kuzik (2021) |
|---|---|---|---|---|---|---|---|---|---|---|---|---|---|---|---|
| | **Title and Abstract** | | | | | | | | | | | | | | |
| 1a | Identification of the type of study in the title | 0 | 0 | 0 | 0 | 1 | 0 | 1 | 0 | 0 | 1 | 1 | 0 | 1 | 0 |
| 1b | Structured summary of objective, methods, results, and conclusions | 1 | 1 | 1 | 1 | 1 | 1 | 1 | 1 | 1 | 1 | 1 | 1 | 1 | 1 |
| | **Introduction** <br> **Background and Objectives** | | | | | | | | | | | | | | |
| 2a | Scientific background and explanation of rationale | 1 | 1 | 1 | 1 | 1 | 1 | 1 | 1 | 1 | 1 | 1 | 1 | 1 | 1 |
| 2b | Specific objectives or hypotheses | 0 | 1 | 1 | 0 | 1 | 1 | 1 | 1 | 1 | 1 | 0 | 1 | 1 | 1 |
| | **Methods** <br> **Participants** | | | | | | | | | | | | | | |
| 3a | Eligibility criteria for participants | 1 | 1 | 1 | 1 | 1 | 1 | 1 | 1 | 1 | 1 | 1 | 1 | 1 | 0 |
| 3b | Settings and locations where the data were collected | 1 | 0 | 0 | 0 | 0 | 1 | 1 | 0 | 1 | 1 | 0 | 1 | 0 | 0 |
| 3c | A table showing baseline demographic characteristics for each group | 0 | 1 | 1 | 1 | 1 | 1 | 1 | 1 | 1 | 1 | 1 | 1 | 1 | 1 |
| | **Sample Size** | | | | | | | | | | | | | | |
| 4a | The sample size has been determined | 1 | 1 | 1 | 1 | 1 | 1 | 1 | 1 | 1 | 1 | 1 | 1 | 1 | 1 |
| 4b | When applicable, explanation of how sample size was determined | 1 | 0 | 1 | 1 | 1 | 1 | 1 | 1 | 0 | 1 | 1 | 1 | 1 | 1 |
| | **Procedure** | | | | | | | | | | | | | | |
| 5a | The procedure has sufficient details to allow replication, including how and when they were actually administered | 1 | 1 | 1 | 1 | 1 | 0 | 1 | 1 | 0 | 1 | 1 | 1 | 1 | 0 |
| | **Instrument or Tools** | | | | | | | | | | | | | | |
| 6a | Completely defined pre-specified primary and secondary outcome measures, including how and when they were assessed | 1 | 1 | 1 | 1 | 1 | 1 | 1 | 1 | 1 | 1 | 1 | 1 | 1 | 1 |
| 6b | Use of validity and reliability tools. | 1 | 1 | 1 | 1 | 1 | 1 | 0 | 1 | 1 | 1 | 1 | 1 | 1 | 1 |
| | **Implementation** | | | | | | | | | | | | | | |
| 7a | Who made each part of study | 0 | 1 | 0 | 0 | 0 | 0 | 1 | 0 | 1 | 1 | 1 | 1 | 1 | 1 |
| | **Statistical Methods** | | | | | | | | | | | | | | |
| 8a | Statistical methods used to analyze the results | 1 | 1 | 1 | 1 | 1 | 1 | 1 | 1 | 1 | 1 | 1 | 1 | 1 | 1 |
| 8b | Use of methods for additional analyses to objective of study | 1 | 1 | 1 | 1 | 1 | 1 | 1 | 1 | 1 | 1 | 1 | 1 | 1 | 1 |
| | **Results** <br> **Outcomes and Estimation** | | | | | | | | | | | | | | |
| 9a | A table or figure showing outputs of analysis more relevant of study | 1 | 1 | 1 | 1 | 1 | 1 | 1 | 1 | 1 | 1 | 1 | 1 | 1 | 1 |
| | **Discussion** <br> **Interpretation** | | | | | | | | | | | | | | |
| 10a | Interpretation consistent with results, balancing benefits and harms, and considering other relevant evidence | 1 | 1 | 1 | 1 | 1 | 1 | 1 | 1 | 1 | 1 | 1 | 1 | 1 | 1 |
| | **Limitations** | | | | | | | | | | | | | | |
| 11a | Study limitations, addressing sources of potential bias, imprecisions, etc. | 1 | 1 | 1 | 1 | 1 | 1 | 1 | 1 | 1 | 1 | 1 | 1 | 1 | 1 |
| | **Practical Implication** | | | | | | | | | | | | | | |
| 12a | Main applicability to results of study | 1 | 1 | 1 | 1 | 1 | 1 | 1 | 1 | 1 | 1 | 1 | 1 | 1 | 1 |
| | **Other Information** <br> **Funding** | | | | | | | | | | | | | | |
| 13a | Sources of funding and other support, role of funders | 1 | 1 | 0 | 0 | 1 | 0 | 1 | 1 | 1 | 1 | 0 | 0 | 1 | 1 |
| | **Total** | 16 | 17 | 16 | 15 | 18 | 16 | 19 | 17 | 17 | 20 | 17 | 18 | 19 | 16 |
| | **Rater agreement (%)** | 90% | 85% | 95% | 90% | 90% | 85% | 90% | 90% | 100% | 100% | 90% | 95% | 95% | 90% |

80], 4 [71], and 6 [74]. Most of the studies utilized multiple data collection tools, indicating potentially higher reliability of the articles.

## Results

### Study flow

Fig 1 shows the flowchart of the current review, illustrating the process from 310 studies to the inclusion of 14 articles in the analysis.

**Thematic analysis.** Selected studies on the relationship between language development, executive function, and screen time are synthesized to explain this complex relationship. Before presenting the identified themes, it is important to define key concepts that recur throughout our analysis.

*Educational media.* It is defined as content specifically designed with learning objectives in mind, including formats such as educational TV, apps, and games. While educational TV typically involves passive viewing, educational apps and games require active engagement from the user, such as completing tasks or interacting with the content to facilitate cognitive development (e.g., Zhang et al. [66]; Hutton et al. [75]).

*Interactive media.* It is noteworthy to distinguish between two types of interactivities; physical and social. Physical interactivity involves the user's direct engagement with the media, such as tapping or swiping, whereas social interactivity entails contingent social responses, either with another person through the media or in reaction to prompts within the media itself (Hutton et al. [75]; Ribner et al. [79]; Zhang et al. [66]).

Following these definitions, the analysis identified the following key themes: (1) screen content and adherence to guidelines, (2) parent-child interaction and family context, (3) passive and active screen time, and (4) attention issues. These themes reflect specific areas where the relationship between screen time and developmental outcomes is most evident and were extracted from the analyses of the reviewed studies. The subsequent section presents each thematic area in detail.

**Screen content and adherence to guidelines.** The nature of screen media and the way children interact with it are of critical importance when considering the effects of children's screen time. Interactive and educational media content has been found to have positive effects

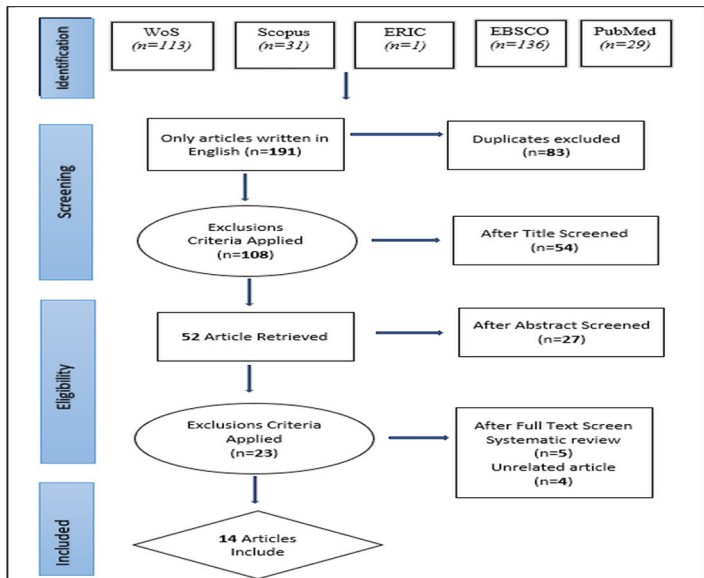

**Fig 1.**

on children's language development and executive functions. Dolgikh et al. [72] investigated the impact of attending extra classes on the development of language development and executive functions with 124 children, with 60 attending extra classes (e.g., music, art, language) twice a week for 4 hours, and 64 not attending any extra classes. In this study, screen time was assessed through a questionnaire administered to the mothers of the participating children. The questionnaire specifically asked about the average amount of screen time the children engaged in per week. The study compared screen time between two groups: children attending extra classes and those who did not. Although the difference in screen time between the groups was not statistically significant, it was noted that children in the extra classes group had slightly lower screen time (approximately 1016 minutes per week) compared to the no extra classes group (approximately 1299 minutes per week).

This screen time data was considered in the analysis of the children's executive functions, ensuring that its potential impact on cognitive development was accounted for in the study's findings.

They observed the positive effects of extra preschool lessons on verbal working memory, highlighting how educational content can support language development. The results indicated that children who participated in extra classes exhibited significantly higher development in verbal working memory over a year compared to those who did not while they report the absence of statistically significant differences between groups who attend extra classes and who do not in terms of executive function development. The difference in the development rate of verbal working memory seems to be directly associated with participation in extra classes. It has been observed that there is a significant positive correlation between active screen time and cognitive development as well as social development. In this study, no specific data was collected regarding the use of screen time for educational purposes during the extra classes. In other words, the relationship between extra classes and screen use was not directly examined. The study concluded that participation in extra classes had a positive impact on the development of children's executive functions. Therefore, the study does not provide definitive conclusions on whether the supplementary classes balance out the negative effects of high screen time. On the other hand, the article did examine screen time as a factor related to the development of children's executive functions, emphasizing that excessive screen time could have negative effects on the development of executive functions. Additionally, the study noted that children not attending extra classes had higher screen time and lower maternal education levels, both of which are known to negatively impact executive function development. These findings suggest that structured extra classes can mitigate some of the negative effects of high screen time, enhancing language development and executive functions in preschool children. These findings indicate that interactive and educational screen content supports children's language development and cognitive skills. Parents' adherence to screen time recommendations also enables beneficial characteristics of screen media which refers to the positive attributes of screen media when used in a controlled and appropriate manner, particularly in alignment with established screen time guidelines, it was linked to better working memory [66]. The "beneficial characteristics" of screen media include educational content and structured use, which support cognitive development when screen time is kept within recommended limits, unlike excessive use that may harm cognitive functions like working memory as stated in the study [66]. The authors investigated the impact of screen time on cognitive development of preschoolers aged 36 to 60 months in Canada by means of a parental questionnaire assessing screen time and cognitive development via Early Years Toolbox. They observed positive effects of screen time in compliance with the recommended guidelines on cognitive skills and language development, particularly noting that educational content and interactive applications support these positive effects. However, they did not find a significant relationship between

screen time and expressive vocabulary in preschool children. The findings suggest that screen time, regardless of its type, may not be related with expressive vocabulary. This implies that children's verbal expression abilities may not be directly affected by screen time. In addition, the relationship between screen time and executive functions, excluding verbal working memory, indicates that both screen time and maternal education level are associated with all executive functions. Controlling screen time and ensuring access to quality content can encourage active participation in learning processes, thus enhancing critical thinking and problem-solving skills. Zhang et al. [66] also emphasize that not only the amount but also the quality of screen time is important. The significance of adhering to screen time guidelines was underscored by Hutton et al. [75] as well. The authors examined the associations between screen media use and the integrity of brain white matter tracts supporting language and literacy skills in preschooler healthy children aged 3 to 5 years in US. They identified a relationship between excessive screen-based media use beyond the guidelines and reduced microstructural integrity of brain white matter tracts that support language and emergent literacy skills in preschool-aged children. Higher screen time scores are associated with higher resistance to diffusion (RD) in white matter bundles related to language, executive function and developing literacy abilities. The findings indicate that higher ScreenQ scores, 15-item measure of screen-based media use, are significantly associated with lower fractional anisotropy (FA) and higher radial diffusivity (RD) in various white matter tracts across the whole brain. These correlations were observed after controlling for child age and household income, and the results remained statistically significant. This was evidenced by lower scores on related cognitive assessments. White matter tracts are known to facilitate communication between nerve cells and form the basis of cognitive functions. Preserving the integrity of these tracts is critical for cognitive development, especially during early childhood. Hutton et al.'s [75] study shows that screen-based media use can positively impact these tracts as long as screen time guidelines are followed and supporting the development of language and literacy skills despite many of the relationships between screen time and children's cognitive development being deemed insignificant.

Screen time's impact manifests simultaneously on both language development and executive functions, with the nature of content playing a crucial role. Dolgikh et al. [72]'s study demonstrated this interconnected relationship: children attending extra classes showed enhanced verbal working memory (a component of both language development and executive functions) despite similar screen time exposure. This finding suggests that structured educational content can support both developmental domains concurrently. Zhang et al. [66] further reinforced this dual impact, showing that adherence to screen time guidelines was associated with improvements in both working memory and language skills, highlighting how appropriate screen use can simultaneously benefit both areas. Hutton et al. [75]'s neuroimaging findings provided biological evidence for this interconnection, showing how excessive screen time affects white matter tracts that support both language processing and executive function development.

**Parent-child interaction and the family context.** Among the examined studies six of them explored the relationship between language development, executive function, and screen time in terms of parent-child interaction from various aspects. Exposure to screen media has been associated with a reduction in the quality of parent-child interactions, particularly when parents do not engage in sufficient conversation with their children. Supanitayanon et al. [80] conducted research to investigate the associations between the age of onset of screen media exposure, cumulative high media exposure, and verbal interaction during the screen time in the first two years of life, and the cognitive development of 4-year-old children with 274 healthy participants by collecting screen media data at specific ages, evaluating cognition at ages 2, 3, and 4 years, while considering parenting behaviors. Findings of this study indicated

that children with earlier screen exposure, more screen time and less verbal interaction during screen time in the first two years of life had lower cognitive and language development in pre-school. Furthermore, it is found out that excessive screen media exposure and lack of verbal interaction were associated with lower fine motor skills and lower visual reception. As children need continuous verbal stimulation to develop their language skills and cognitive abilities, increased verbal interactions between parents and children can support these developmental processes. In a similar vein, Carson and Kuzik [70] examined the association between parent-child interference across various devices and measured cognitive development and social-emotional development in preschool-aged children with 100 participants in Canada. They emphasized potential risks and negative impact of electronic devices on early childhood development, parent-child technology interference could interrupt parent-child interactions during the day. Higher parent-child interference was associated with lower response inhibition and emotional self-regulation. To illustrate, when children seek emotional support and guidance, they may be unable to meet these needs due to parental distractions. These findings indicate that parents should regulate their technology use besides from children's screen time and make it interactive.

Cultural and socioeconomic factors have been found to be critical factors that affect parent-child interaction. Three studies measured how those factors could influence parent-child inter-action when they examine the relationship between language development, executive func-tions, and screen time. Studying the role of cultural and socioeconomic factors on the relationship between language development, executive functions and screen time, Zhang et al. [81] stated that the effects of screen time are largely shaped by several factors, including the life conditions of children and the educational levels of their parents. For instance, in families with higher socioeconomic status, children often have access to more educational resources and parents are better able to manage their children's screen time. This can mitigate the adverse effects of screen time and facilitate children's cognitive and language development. The authors found out that children who adhered to the screen time guidelines showed greater improvement in intellectual skills over time, especially in language development and vocabu-lary knowledge, compared to those who did not follow the screen time recommendations. Medawar et al. [77] also reached comparable conclusions in their study, which examined the effect of mothers' home literacy beliefs and practices and the quantity and quality of screen media exposure to Argentinian children aged 18 months to 36 months. Their findings pointed out that higher parental education levels were associated with better language development in children, literacy beliefs and practices, particularly those of mothers, were significantly associ-ated with better language outcomes in children. The findings of the study suggest that the qual-ity of parent-child verbal interactions during shared media use contributes significantly to language development by creating an opportunity for dialog and vocabulary learning. The type and quality of screen media exposure were found to have different impacts on language development. Higher SES families often had better access to high-quality educational content and engaged more in joint media engagement, which positively influenced language outcomes. Conversely, passive screen exposure (e.g., TV) was more prevalent in lower SES families and was negatively associated with language development. Lastly, they noted that higher SES parents were more likely to engage in verbal scaffolding and joint media engagement with their children, which further supported language development. Besides, Hu et al. [74] also emphasized that parent-child interaction plays a pivotal role in enhancing language develop-ment through shared media usage, as suggested by their findings. In families with a lower socioeconomic status, parents may frequently be required to work longer hours, which may result in a reduction in the quality of time they are able to spend with their children. This can have a negative impact on children's language development. Furthermore, these families may

be constrained in their ability and opportunity to access educational content, which can impede children's learning opportunities. It is possible that families with low socioeconomic status may utilize screen time as a form of childcare, which could result in increased screen time and negative effects. In parallel with these two studies, Hendry et al. [73] observed a consistent negative association between low socioeconomic status and screen time. Furthermore, the authors proposed that these families may be more vulnerable to the adverse effects of prolonged screen exposure.

The relationship between screen time, language development, and executive functions is significantly mediated through parent-child interactions. Supanitayanon et al. [80] demonstrated how early screen exposure combined with reduced verbal interaction impacts both cognitive domains: children with earlier screen exposure and less verbal interaction showed lower performance in both language skills and cognitive functions. Carson and Kuzik [70] extended this understanding by showing how parent-child technology interference simultaneously affects response inhibition (an executive function) and emotional self-regulation, which in turn influences language learning opportunities. The socioeconomic context further shapes this three-way relationship. Zhang et al. [81] found that higher socioeconomic status families could better manage screen time while providing educational resources that support both language and cognitive development. Medawar et al. [77]'s findings reinforced this, showing how maternal literacy beliefs and practices influence both language outcomes and executive functioning through quality screen time management. Hu et al. [74] and Hendry et al. [73] further demonstrated how socioeconomic status affects both developmental domains through screen time exposure patterns and parent-child interaction quality.

**Passive and active screen time.** In examining the relationship between language development, executive functions, and screen time, it is essential to consider the way children are exposed to screens. Four of the studies focused on how children were exposed to screen media. Medawar et al. [77] figured out that passive screen exposure (TV viewing) has a detrimental impact on language and literacy abilities, and that these effects are linked to self-regulation challenges in their study, which involved 465 mothers of Argentinian children aged 18–36 months. They reported that the continuous presence of television in the background can act as a distraction, impeding the development of language and literacy skills in children. A constant stream of auditory or visual stimuli in the background can impede children's ability to maintain attention and concentration, thereby negatively affecting their learning process. Self-regulation difficulties can impede children's capacity to focus their attention, regulate their emotions, and control their behavior, which can negatively impact their academic and social achievement. Another study conducted by Hu et al. [74] investigates the role of active and passive screen time on Chinese children's social and cognitive development with 579 children aged 5. They found that passive screen time (TV viewing) is significantly and negatively associated with children's executive functioning and social skills while there exists a meaningful and positive correlation between children's active screen time and receptive language skills. Executive functions encompass a range of higher-order cognitive abilities, including planning, problem solving, attention control, and flexibility. The development of these critical skills may be inhibited by passive screen time. Furthermore, television viewing may impede children's opportunities for social interaction, thereby hindering their capacity to develop social skills. It is evident that children require face-to-face interactions to develop their social skills. Therefore, based on the findings of study, it could be concluded that television viewing cannot replace these interactions and may prevent children from developing skills such as empathy, emotional expression, and understanding social cues.

Veraksa et al. [19] demonstrated that passive screen time, particularly television viewing, is negatively associated with children's ability to process verbal information effectively,

development of executive functions, and social skills in their study conducted with 122 preschool children aged 5–6 in Russia whereas they noted. They found that passive screen time has negative effect on preschoolers' phonological memory development, especially for those who engage in longer daily periods of television viewing. The capacity to process verbal information is of critical importance for children's ability to learn and comprehend language; therefore, it could be argued that television viewing or passive engagement with digital devices may prevent children from using and developing these skills based these findings. Another study that showed the negative effects of passive screen time was conducted by Kim and Chung [76]. To investigate the association between exposure to TV or video and children's language development and school achievement in childhood, Kim and Chung [76] conducted annual assessments from birth to 87.9 months, providing a comprehensive analysis of the longitudinal impact of screen time on child development. They have emphasized the negative effects of early screen exposure on language development. Their research shows that prolonged television exposure at age two negatively impacts children's language development and school achievement. At this age, children lay the foundations of their language skills and are greatly influenced by the speech and interactions in their environment. Excessive television use can reduce these critical interactions, adversely affecting children's language acquisition processes, and passive screen time can limit children's active participation and interaction, restricting the natural learning opportunities that support language development.

Furthermore, passive screen time can impede the interactions that facilitate children's vocabulary and language comprehension development. Constant background television can weaken children's ability to focus, negatively impacting their capacity for deep thinking and complex problem-solving. In line with these studies, Ribner et al. [79] conducted a nationally representative study with children ages from 3 to 7 in USA which aims to investigate the effect of the foreground TV exposure (educational, entertainment television), and background TV on self-regulation problems, language, and literacy skills. They observed that background and entertainment TV were negatively associated with language and literacy skills while educational TV was not. Furthermore, self-regulation problems might serve as a mechanism linking TV exposure to language and literacy skills. Likewise self-regulation problems, emotional lability is one aspect that has been found to have a negative relationship between screen time. Oflu et al. [78] observed that prolonged screen time can lead to emotional lability and attention problems in children in their study which examines excessive screen time associated with emotional lability in preschool children aged 2 to 5 years in Türkiye. As screen time increases, children's ability to control and regulate their emotional responses may weaken. This can lead to negative outcomes in children's social relationships and school performance. To illustrate, children with poor emotional regulation skills may struggle to concentrate in the classroom and complete academic tasks. In addition, Kim and Chung [76] highlighted the negative effects of screen time on self-regulation skills in early childhood. Self-regulation skills encompass the ability to control one's thoughts, emotions, and behaviors. The development of these skills in early childhood is critical for academic and social success later in life. Prolonged screen time can hinder children's development of these skills and weaken their ability to control themselves. These findings indicate that screen time can have negative effects on children's emotional regulation skills, which can indirectly impact language development and other cognitive functions. It is important to carefully manage screen time and support children's self-regulation skills. The use of passive screen time can impede the development of critical cognitive and language skills in children, including attention, executive functioning, and social skills. Nevertheless, the promotion of active and interactive screen content may serve to mitigate these negative effects.

The distinction between passive and active screen time reveals crucial differences in their impact on both language development and executive functions. Medawar et al. [77] found that passive screen exposure negatively affects both language acquisition and self-regulation abilities. This finding was corroborated by Hu et al. [74], who showed that while passive screen time negatively impacted executive functioning and social skills, active screen time positively correlated with receptive language skills, demonstrating the differential effects of screen engagement types on both developmental domains. Veraksa et al. [19] provided further evidence of this interconnection, showing how passive screen time simultaneously affects verbal information processing and executive functions. Kim and Chung [76]'s longitudinal study demonstrated how early screen exposure impacts both language development and academic achievement through shared cognitive mechanisms. Ribner et al. [79] extended this understanding by showing how background and entertainment TV affect both language and literacy skills through self-regulation pathways. Oflu et al. [78] and Kim and Chung [76] highlighted how emotional regulation, influenced by screen time, acts as a bridge between language development and executive functioning.

**Attention issues.**   Excessive screen time can negatively impact children's working memory, executive functions, and language development. Therefore, it emphasizes the importance of high-quality content and adherence to recommended guidelines. Zhang et al. [81] and Hendry et al. [73] emphasize the negative effects of total screen time, particularly television watching, on children's working memory and executive functions. Working memory is defined as the capacity to store and process information in the short term, playing a critical role in children's learning and problem-solving abilities. Zhang et al. [81] investigated longitudinal associations of subjectively measured physical activity and screen time with multiple domains of cognitive development in children aged 2.5 to 5 years old and excluded the ones who have conditions limiting physical activity or affecting cognitive performance. They figured out that using screen media in accordance with guidelines is associated with better intellectual ability and language development. Besides, Hendry et al. [71] investigated how variability in the home environment before and during the 2020 pandemic in the United Kingdom, involving 575 children aged 8 to 36 months in a parent-reported study. The authors noted that not only the amount but also the content of screen time has a significant impact on children's attention and focus skills. This study found that higher screen time in early childhood was associated with lower cognitive outcomes, specifically in executive function and regulation skills. The findings of this theme showed that screen use, particularly in infants, negatively affected cognitive executive function (CEF) and regulation, and it mediated the relationship between socio-economic status (SES) and these cognitive outcomes. Specifically, fast-paced and distracting content can shorten children's attention spans and weaken their ability to concentrate. The findings suggest that exposure to such content may lead children to seek similar levels of constant stimulation in real life, making it difficult for them to maintain focus in classroom settings and other learning environments. It is shown that with increased screen time, a significant decline in children's working memory capacity has been observed. This decline can lead to increased forgetfulness in daily life, difficulties in learning new information, and challenges in completing complex tasks.

The impact of screen time on attention serves as a crucial link between language development and executive functions. Zhang et al. [81] and Hendry et al. [73] demonstrated how screen time affects working memory and executive functions, which in turn influence language processing and acquisition. Their research showed that adherence to screen time guidelines benefits both intellectual ability and language development through shared attention and cognitive control mechanisms. Specifically, Hendry et al. [74] found that excessive screen time negatively impacts cognitive executive function (CEF) and regulation, which serve as

foundational skills for both language processing and executive control. The research highlighted how attention capacity, affected by screen exposure, serves as a common pathway influencing both language acquisition and executive function development.

## Discussion

The purpose of this study was to investigate the interconnected relationship between language development, executive function, and screen time in children aged 0–78 months through a systematic review. This relationship is particularly complex during this developmental period, as executive functions and language development are mutually influential- executive functions support language acquisition while language skills facilitate executive control development. Given the pervasiveness of digital tools in contemporary society, understanding how screen time affects both these developmental domains simultaneously is crucial for this age group. Chan & Rao [83] and Cumming et al. [84] emphasize this interconnection, noting how executive functions and language development co-develop and mutually influence each other during early childhood (0–78 months). This relationship has been extensively studied in both typically developing children [85] and those with developmental challenges [86–89], revealing how executive functions support various language skills including vocabulary acquisition [90], new word learning [91], and reading comprehension [92–96] while language skills in turn scaffold executive control development.

The gaps identified by Filipe et al. [97] and Hartanto et al. [98] regarding specific cognitive-language relationships and executive function-screen time connections in children aged 0–78 months underscore the importance of examining these domains together. Previous reviews like Karani et al. [49] and Bustamante et al. [53] have examined these relationships separately, finding negative impacts of prolonged screen time on language development but inconclusive results for executive functions in this age group. Our study uniquely contributes to literature by examining how screen time simultaneously affects both domains during this critical period (0–78 months), building on Streegan et al.'s [99] work with older children.

Our thematic analysis revealed four interconnected themes that demonstrate how screen time influences both language development and executive functions in children aged 0–78 months. Regarding screen content and adherence to guidelines, we found that educational content can simultaneously support both domains during this developmental period- enhancing language skills while developing executive control through structured learning experiences. The quality of interaction during screen time emerged as crucial for this age group, as parental involvement creates opportunities for both language learning and executive function development through guided engagement.

The role of parent-child interaction and family context revealed how screen time's impact on both developmental domains is mediated through social interaction in children aged 0–78 months. Two critical findings emerged: First, insufficient parent-child interaction during screen time impairs both language development and executive control processes that rely on social scaffolding at this age. The literature emphasizes parents' crucial role in mitigating negative effects of screen exposure on both domains [100–102]. Wong et al. [103] extended this understanding by showing how parents' own device use can disrupt parent-child interaction during these crucial early years, negatively affecting both language learning opportunities and executive function development. This finding particularly matters since quality interaction supports both domains simultaneously during this period. Second, socioeconomic factors influence access to educational resources and parents' ability to manage screen time, affecting both domains simultaneously in children aged 0–78 months. Studies like Slobodin et al. [104]

and Hoffman et al. [105] support these findings, showing how socioeconomic status moderates screen time's impact on both language and cognitive development during early childhood.

The distinction between passive and active screen time provided crucial insights into how different types of engagement affect both developmental domains in children aged 0–78 months. Passive screen exposure negatively impacts both language processing and executive functions, particularly attention and social skills [106–108] during this period. Conversely, active screen time with educational and interactive content can positively influence both domains [109–112], as demonstrated by studies showing improvements in both vocabulary and executive function skills [51, 52, 113] in this age group.

The attention issues theme highlighted how screen time affects cognitive processes fundamental to both language development and executive functions during this critical period (0–78 months). Excessive screen time impairs attention control in young children, which simultaneously affects language learning processes and broadens executive functions. This creates a cascade effect where attention deficits disrupt both vocabulary acquisition and cognitive control development during these formative years. Studies by Kebir and Özkaya [114] and Schwarzer et al. This interconnected impact was supported by [115], showing how excessive screen time simultaneously affects cognitive, language, and socio-emotional skills in children aged 0–78 months.

The diversity in findings across studies reflects the complex nature of how screen time influences both developmental domains during early childhood (0–78 months). Factors such as content interactivity, parental involvement, and developmental timing all contribute to whether screen time supports or hinders the co-development of language and executive functions at this age. This complexity is further illustrated by contrasting findings from studies such as Taylor et al. [116] and Dore et al. [117], who found no significant relationships in this age group, with other studies showing clear impacts on both domains.

Our findings emphasize the need to consider language development and executive functions as interrelated processes when studying screen time's impact on children aged 0–78 months. Future research should focus on understanding how different types of screen engagement can be optimized to support both domains simultaneously during this crucial developmental period. Additionally, more attention should be paid to how socioeconomic and cultural factors moderate screen time's impact on these interconnected developmental processes in early childhood.

## Limitation

Several limitations were considered when interpreting the findings of this study. The systematic review was limited to the Web of Science, Scopus, ERIC, EBSCO, and Pub-Med databases and selected keywords in the search process. While these databases were chosen in line with the research's purpose, this limitation may have affected the comprehensiveness of the study. In addition, the analysis was conducted with the articles written in English-language only, which represents another limitation to be considered. Excluding various scientific documents may lead to an incomplete interpretation of the literature. It is important to recognize that the assessments of the quality of reporting of studies based on the findings of this systematic analysis may be constrained by the limitations identified within the analysis itself. The sample size was limited to 14 articles, which may limit the generalizability of the results and may not fully encompass the entire body of literature in this field.

## Conclusion

Our study's findings provide valuable insights into the complex relationships between children's language development, executive functions, and screen time. It emphasizes the importance of considering these three variables together, particularly in the preschool period. The results of this study are consistent with previous research indicating that educational and interactive screen content has a positive impact on children's cognitive and language development. Excessive and passive screen use has been shown to have a negative effect. Additionally, the study demonstrates that parent-child interaction and socioeconomic factors play a significant role in this relationship. These findings are expected to inform important strategies for parents, educators, and policymakers in managing and guiding children's screen use. In conclusion, the quality of time children spends in front of screens and the way parents manage this process have a significant impact on language development and executive functions.

## Implications

The implications of these findings are significant for parents, educators, and policymakers. For parents, it is crucial to carefully select interactive and educational screen content to support children's language and cognitive development. Adherence to recommended screen time guidelines can minimize potential negative effects on executive functions and language skills. Engaging in screen media activities with children enhances interaction and supports developmental processes. Limiting passive screen time and ensuring that screen use is balanced with real-world interactions and play opportunities are also essential. For educators, integrating educational and interactive screen content into the curriculum can support language and cognitive skills development. Educating parents on the importance of managing screen time and selecting high-quality content is also crucial. Additionally, promoting activities that reduce passive screen time and encourage active learning and interaction among children is beneficial. For policymakers, it is important to develop and promote guidelines for screen time that emphasize the importance of content quality and interactive use. Supporting initiatives that provide access to high-quality educational media for families from all socioeconomic backgrounds and implementing programs that educate parents about the impact of screen time on child development and strategies for managing screen media use effectively are also key steps. By addressing these areas, it is possible to harness the potential benefits of screen media while mitigating its negative impacts on children's development. Carefully managing screen time not only supports children's language and cognitive development but also helps them develop healthy digital habits. Parents can teach their children how to use digital media safely and effectively, improving their digital literacy skills, which is a critical skill for future success in the digital world. In an era where digital technology is becoming increasingly prevalent, it is of paramount importance to gain a deeper understanding of the relationship between screen time, language development, and executive function. Consequently, it is recommended that future systematic reviews expand their research scope to include methodological quality.

## Supporting information

**S1 File. Dataset.** This Excel file contains the data used in this study, including all variables and analyses reported in the manuscript. It is publicly available and can be accessed directly from the OSF repository.
(XLSX)

**S2 File. PRISMA checklist.** The PRISMA checklist outlines the systematic review process and compliance with reporting standards.
(PDF)

**S3 File. Handling missing data.** This document outlines the methodological approach used to handle missing data in the systematic review, ensuring rigor and comprehensiveness at every stage of the analysis.
(DOCX)

**S1 Table. Evaluation process of studies identified in systematic review.** This table summarizes the evaluation process of studies identified in the systematic review.
(DOCX)

**S2 Table. Data extraction summary.** This table provides details on the data extracted from the included studies.
(DOCX)

**S3 Table. Risk of bias and quality/certainty assessments.** This table presents the GRADE assessment of included studies.
(DOCX)

## Author Contributions

**Conceptualization:** Ayşe Gül Kara Aydemir, Gülüzar Şule Tepetaş Cengiz, Ahmet Altındağ.

**Data curation:** Mazhar Bal, Ayşe Gül Kara Aydemir.

**Formal analysis:** Mazhar Bal, Ayşe Gül Kara Aydemir.

**Investigation:** Gülüzar Şule Tepetaş Cengiz, Ahmet Altındağ.

**Methodology:** Mazhar Bal, Ayşe Gül Kara Aydemir.

**Project administration:** Mazhar Bal, Ayşe Gül Kara Aydemir, Gülüzar Şule Tepetaş Cengiz.

**Resources:** Mazhar Bal.

**Supervision:** Gülüzar Şule Tepetaş Cengiz.

**Validation:** Mazhar Bal, Ayşe Gül Kara Aydemir, Gülüzar Şule Tepetaş Cengiz.

**Writing – original draft:** Mazhar Bal, Ayşe Gül Kara Aydemir, Gülüzar Şule Tepetaş Cengiz, Ahmet Altındağ.

**Writing – review & editing:** Mazhar Bal, Ayşe Gül Kara Aydemir, Gülüzar Şule Tepetaş Cengiz, Ahmet Altındağ.

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
