## [Decision Letter · Decision Letter 0]

17 May 2024

PONE-D-24-04813Examining the Relationship Between Language Development, Executive Function, and Screen Time: A Systematic ReviewPLOS ONE

Dear Dr. Kara Aydemir,

Thank you for submitting your manuscript to PLOS ONE. After careful consideration, we feel that it has merit but does not fully meet PLOS ONE’s publication criteria as it currently stands. Therefore, we invite you to submit a revised version of the manuscript that addresses the points raised during the review process. Please pay especially close attention to the issues raised by Reviewer 1 regarding the overall quality of your study and manuscript to adhere to scientific standards and quality criteria for systematic reviews. There are major concerns addressed by both reviewers regarding the general methodological quality and rigor that will be crucial concerning the further development of your manuscript and its potential for acceptance for publication. You will find the reviewers' comments in the body of this email and as detailed remarks from reviewer 1 attached below.

We look forward to receiving your revised manuscript.

Kind regards,

Alexander Röhm

Guest Editor

PLOS ONE

Journal Requirements:

Reviewers' comments:

Reviewer's Responses to Questions

**Comments to the Author**

1. Is the manuscript technically sound, and do the data support the conclusions?

Reviewer #1: No

Reviewer #2: Partly

2. Has the statistical analysis been performed appropriately and rigorously? 

Reviewer #1: No

Reviewer #2: No

3. Have the authors made all data underlying the findings in their manuscript fully available?

Reviewer #1: No

Reviewer #2: No

4. Is the manuscript presented in an intelligible fashion and written in standard English?

Reviewer #1: No

Reviewer #2: Yes

5. Review Comments to the Author

Reviewer #1: The recommendation of a major revision has been given due to major corrections that have to be done. Besides plenty errors and inaccuracies in the methodology, that make the review unreproducible and intransparent, these mainly concern the wording and the text structure. Especially the sections results, discussion, and implications have to be revisioned regarding the wording. Writing a systematic review requires a range of skills. It involves thorough literature searching, critical evaluation, and clear presentation of the results. The writer's effort is commendable for attempting this complex task. Nevertheless, the text still needs some polishing (a few rounds of feedback and proofreading) to make it publishable. Otherwise it does not meet the criteria of good scientific practice

Reviewer #2: Thank you for the opportunity to review this article, Examining the relationship between language development executive function and screen time – a systematic review. This is a growing area of importance for policy makers, practitioners and families and so it is very valuable to have systematic reviews to consolidate the research.

While I applaud the authors for this comprehensive review, there are some key methodological concerns that should be addressed prior to publication.

Critique of previous systematic reviews:

There are several recent systematic reviews that have been published in this area (or very closely aligned areas). It would be useful to see a description of these reviews, what their limitations or gaps are and how this current review addresses them. This can then be reflected on in the discussion in the ways previous research has been limited or inconclusive and what the current findings have added. Some examples of the reviews include:

o Madigan, McArthur, Anhorn et al (2020) Associations between screen use and child language skills: a systematic review and meta-analysis

o Streegan, Lugue & Morato-Espino (2022) Effects of screen time on the development of children under 9 years old: a systematic review

o Santos, Mendes, Miranda & Romano-Silva (2022) The association between screen time and attention in children: a systematic review

Description of age ranges:

The authors should provided a clearer description within both the methods and results of what they mean by “preschool children” and “early childhood”. This can mean different things in different contexts and so it is important to be clear on the age range. Of the included studies the age ranges should also be included for all studies listed in Table 1. Within the results section I noted, two studies appear to fall outside the typical early childhood age range and so the authors should make clearer the justification of their inclusion. Specifically:

- Vohr et al focused on “children born preterm” however the study was conducted when the child was 6-7 years old.

- Horowitz-Kraus & Hutton focused on children 8-12 years old.

Methodological quality review:

The authors have used the CONSORT guidelines to support a review of methodological quality. However, I would argue this focuses on more on the quality of reporting the studies and not necessarily the methodological quality. Typically some form of “Risk of Bias” assessment would be used to examine methodological quality there are several tools available depending on the study design. An article by Ma, Wang, Yang et al (2020) “Methodological quality (risk of bias) assessment tools for primary and secondary medical studies: what are they and which is better” has a useful summary of the available tools.

Furthermore, on the use of CONSORT there are different guidelines depending on study design but it is unclear if the authors have used these variations depending on the study design on their 16 articles included in the review. I would have also hoped to see an integration of the thematic review and the methodological quality to inform conclusions regarding the relationship between screen time, language development and executive functioning. For example, the authors ultimately conclude (pg 17) that “there is no relationship between executive function and language development, regardless of the type of screen time”. However, this does not seem to be based on an assessment of the quality of these studies that reported no association (or at least this is not detailed in the paper).

Screen time x Language Development x Executive Functioning:

After reading the paper it is not clear to me if studies in the review had to include data on all three constructs or if eligible papers may have included data only on ‘screen time and language development’ and ‘screen time and executive functioning’. Providing this detail would be useful, including detailing outcome measures in Table 1 and providing a more nuanced discussion of differences in findings between language development and executive function.

Within the discussion, the authors also state “Despite advances in understanding the link between language and executive function, the current understating of the relationship between specific cognitive function components and language skills is still limited”. However, it does not appear this was a question the authors were trying to understand in their review – given all the papers were in the context of screen time.

Multiple data collections tools:

It is not clear what the authors are referring to when using this term. Is it related to studies using self-reported vs objective measures of the constructs or that the measure of screen time includes multiple types of devices? If it is the latter, this is an interesting point however, it is not clear in the results as to whether this has made a difference in the conclusions. This may also be a result of authors not restricting date ranges in their review and so many older studies may be focused entirely on TV viewing as mobile technology was not commonly available in studies published more than a decade ago.

Minor formatting recommendations:

Including the definition of different executive functioning process in a Table might be useful to a reader. You could also include an overall definition for executive function, and any key language development measures you are using so that readers can easily refer back to them if need be.

In the section ‘Thematic analysis’ the authors have not included the details of the ‘non-association’ theme.

This sentence in the discussion may need to be reworked: “The study suggests that early childhood teachers are not sufficiently aware of the importance of executive functions in preschool language education [88]. may lack knowledge.”

There are several hyphens used throughout the manuscript that should be removed. Some examples include: “de-velopmental” and “emo-tional” in first paragraph of introduction.

6. PLOS authors have the option to publish the peer review history of their article (what does this mean?). If published, this will include your full peer review and any attached files.

Reviewer #1: No

Reviewer #2: **Yes: **Dr Mary Brushe

---

## [Author Response · Author response to Decision Letter 0]

28 May 2024

Sayın Dr. Alexander Röhm,

Mazhar Bal, Ayşe Gül Kara Aydemir, Gülizar Şule Tepetaş Cengiz ve Ahmet Altındağ tarafından hazırlanan “Dil Gelişimi, Yürütücü İşlev ve Ekran Süresi Arasındaki İlişkinin İncelenmesi: Sistematik Bir İnceleme” başlıklı makalemizin gözden geçirilmiş halini PLOS ONE'da yayınlanmak üzere yeniden değerlendirmeye sunmak için yazıyorum. Hakemlerin yorum ve önerilerini dikkatle değerlendirdik ve revizyonların makalemizi önemli ölçüde güçlendirdiğine inanıyoruz. Hakemlerin ve yayın kurulunun makalemizi değerlendirmek için harcadıkları zaman ve çabaya derinden minnettarız.

Makaleyi açıklık ve tutarlılık açısından gözden geçirdik, net olmayan cümleleri yeniden ifade ettik ve belgenin genel okunabilirliğini sağladık. Giriş bölümünde, araştırma boşluklarını vurgulayarak inceleme için ayrıntılı bir gerekçe ekledik. Uyarlanmış CONSORT protokollerinin kullanımına açıklık getirdiğimiz ve dahil etme kriterlerini belirttiğimiz yöntemler bölümünde önemli revizyonlar yapıldı. Veriler yeniden analiz edildi ve tematik kümeler tutarlılığı ve derinliği artırmak için yeniden yapılandırıldı. Sonuç olarak, tartışma bölümü yeniden yazılarak ilgili literatürden ek bilgiler eklenmiş ve bulguların daha yapılandırılmış bir sunumu sağlanmıştır. Tablolar, katılımcı özellikleri ve yaş aralıkları hakkında ayrıntılı bilgi içerecek şekilde güncellenmiştir. Sonuç ve çıkarımlar bölümleri revize edilen içerikle uyumlu olacak şekilde yeniden yazılmıştır. Her bir hakemin yorumlarına verilen ayrıntılı yanıtlar yanıt belgesinde yer almaktadır.

Gözden geçirilmiş makalenin PLOS ONE standartlarını artık daha iyi karşıladığına ve hakemlerin tüm endişelerini kapsamlı bir şekilde ele aldığına inanıyoruz. Gözden geçirilmiş başvurumuzu olumlu değerlendirmenizi bekliyoruz.

Zaman ayırdığınız ve yeniden değerlendirdiğiniz için teşekkür ediyor ve çalışmamızın PLOS ONE'da yayınlanmak üzere yeniden değerlendirilme fırsatını dört gözle bekliyorum. Ayşe Gül Kara Aydemir bu çalışmanın sorumlu yazarıdır. Herhangi bir sorunuz olması veya daha fazla bilgiye ihtiyaç duymanız halinde, lütfen benimle agkara@gmail.com adresinden e-posta yoluyla iletişime geçin.

---

## [Decision Letter · Decision Letter 1]

27 Aug 2024

PONE-D-24-04813R1Examining the Relationship Between Language Development, Executive Function, and Screen Time: A Systematic ReviewPLOS ONE

Dear Dr. Kara Aydemir,

Thank you for submitting your manuscript to PLOS ONE. After careful consideration, we feel that it has merit but does not fully meet PLOS ONE’s publication criteria as it currently stands. Therefore, we invite you to submit a revised version of the manuscript that addresses the points raised during the review process.

Please see the Editor's and the Reviewers' Comments below.

We look forward to receiving your revised manuscript.

Kind regards,

Alexander Röhm

Guest Editor

PLOS ONE

Additional Editor Comments:

Dear Authors,

thank you for your thourough revision of your manuscript. The reviewers very much appreciate the efforts you put into your review and acknowledge the improvements it made since its original submission. However, there are still some minor, but also major issues that deserve your full attention. In line with both reviews I recommend to especially pay close attention to the following points (as well as the reviewers detailed comments):

- Reference and discussion of other recent SR/MA in the field (see Reviewer 3) to highlight the relevance of your specific SR.

- While you elaborate in detail on various aspects surounding language development like executive functions, working memoriy, problem solving, processing etc. the link between these and "screen time" needs to be pointed out more clearly. At the moment, everything is listed rather additive but not put together into a strong line of argumentation and research rationale. Also, please try to avoid redudant argumentation in the regard, for instance, concerning the relation between language development, EF, and WM. What exactly is the important point here for your review?

- The overall aim of the SR is well described. Yet, the research questions do not seem to fully address what is (a) plannend to do and is (b) actually done by the SR. RQ1 could be omitted, since it appears to aim for a assesment of potential biases, which should be added to every SR following the PRISMA guidelines. RQ2 should be formulated as closely as possible to cover the scope of the SR, e.g.: "What is the relationship between children's screen time, language development, and executive functions reported in the present research literature?"

I hope these comments along with the reviewers' comments help to further improve your manuscript.

Best regards.

Reviewers' comments:

Reviewer's Responses to Questions

**Comments to the Author**

1. If the authors have adequately addressed your comments raised in a previous round of review and you feel that this manuscript is now acceptable for publication, you may indicate that here to bypass the “Comments to the Author” section, enter your conflict of interest statement in the “Confidential to Editor” section, and submit your "Accept" recommendation.

Reviewer #2: (No Response)

Reviewer #3: (No Response)

2. Is the manuscript technically sound, and do the data support the conclusions?

Reviewer #2: Partly

Reviewer #3: Partly

3. Has the statistical analysis been performed appropriately and rigorously? 

Reviewer #2: Yes

Reviewer #3: N/A

4. Have the authors made all data underlying the findings in their manuscript fully available?

Reviewer #2: Yes

Reviewer #3: Yes

5. Is the manuscript presented in an intelligible fashion and written in standard English?

Reviewer #2: No

Reviewer #3: Yes

6. Review Comments to the Author

Reviewer #2: Thank you to the authors for their thorough review of the manuscript. I was pleased to see considerable improvements in this version and acknowledge the time and effort that would have gone into the updates to the results and discussion section.

However, these changes have raised some additional concerns that I believe need to be addressed before publication.

Some are minor formatting/grammatical issues and others have larger implications for the overall review findings and methodology. I have attempted to summarise this based on each section of the manuscript:

Abstract:

- “The study categorizes the impact into four key themes…” – it is not clear what ‘impact’ is referring to. I’m assuming this refers to the impact of screen time on language development and executive functioning, but this should be specified.

- The sentence ‘The prevalence of attention problems is found to be higher in individuals who engage in high levels of screen time’ appears out of place and could be removed from the abstract or included earlier on before discussing socioeconomic and cultural factors.

Introduction:

- Overall, the introduction has been significantly improved and the authors should be commended on their excellent summary of the literature. The summary makes a clear case for RQ2 however, the authors have not really identified why RQ1 is needed as a separate aim. Is there previous evidence to say that quality of reporting in this field is poor and so that’s why we need it to be reviewed, or should this be a sub-aim that can be commented on within findings from RQ2? I would recommend listing RQ2 as the primary question and provide greater rationale on why RQ1 is needed.

Some minor edits to this section also include:

o The summary of the link between language development and executive function on page 5 could be shortened to streamline this section.

o Pg 8: “Mantilla and Edwards [51] noted that digital technology is now become and accepted an integral part…” should read “Mantilla and Edwards [51] noted that digital technology has now become and is accepted as an integral part…”

o Noting the authors advised the incorrect placement of hyphens seems to be an issue due to the submission system, however I note one is still present on page 9, end of first paragraph ‘for-mulated’.

Methods

- The inclusion of Table 2 is very helpful and provides a good overview of the included studies. It does say in text that ‘author and year’ is described, however, currently the table only has the authors listed. Including details of the year of publication would be important especially given there was no restriction on publication date.

Results

- Please check the formatting of Table 3: (1) some of the study details in the first row are cut off; (2) the columns have not always been merged correctly for rows where headings are listed; (3) spelling error for 12a heading ‘Practisal’; (4) Rater agreement (%) should have the percentage sign after the number eg., 90%.

- Second sentence in the first paragraph on pg 22, should be reframed to state “Based on these data, all the articles have a value equal to or greater than 15 points.

- Can the authors please clarify how screen time was incorporated into the Dolgikh et al [62] study? Did the extra classes that were attended in the study use educational screen time? The interpretation of this study may also need clarification, as the authors state ‘These findings suggest that structured extra classes can mitigate some of the negative effects of high screen time, enhancing language development and executive functions in preschool children’. However, it is not clear if that was actually tested in this research and is contradictory to the authors earlier statement that attending extra classes had no relationship to executive functioning?

- “Parents adherence to screen time recommendations also enables beneficial characteristics of screen media, it was linked to better working memory” – what do the authors mean by ‘beneficial characteristics’?

- I would recommend the authors rethink the wording of the theme ‘interactive and educational screen content’ to incorporate something about adherence to guidelines as some of the studies (e.g., Hutton et al) does not appear to investigate interactive or educational content specifically. Similarly, I would recommend reframing the theme ‘Parent and child interaction’ to ‘Parent-child interaction and the family context’ given much of the research summarised is focusing on socioeconomic status of the family.

- First paragraph of pg 25 should be included in the Discussion section of the paper, rather than the results and it should also include references to some of the claims made (e.g., interactions can support children’s social and emotional development).

- As described on pg 29, the study by Kim and Chung [66] appear to include children outside the age range of the eligibility criteria (87.9months). Similarly for the study by Ribner et al [69] is outside the age range (7 years).

- Several statements in the second half of pg31 require associated references to support these claims.

- Final paragraph of the results section (pg 31-32) belongs within the discussion section.

Discussion

- The statement on pg 34 “… the results regarding the quality of reporting can inform researchers about the effective methods for data collection and analysis.” I don’t believe is correct. The review of adherence to reporting guidelines makes no comment on the effectiveness of methodology or data analysis. I would further argue that RQ1 does not make a substantial contribution to the existing body of literature and as a reader it is not clear what can be taken from the findings, other than research in this space seems to be generally applying the CONSORT checklist.

- It would be helpful if the authors could make some comment on why there might be diversity in findings across studies within specific themes. For example, they describe that some studies find negative relationships and others found no relationship – but it is unclear why this may be the case. Are there differences in how they defined screentime or the age of the children they assessed?

- I would recommend reframing the sentence “The literature also emphasizes that one of the significant variables affecting the negative impact of screen exposure is parents.” There is considerable context and factors outside of many parents control that may or may not influence their ability to support healthy screen time, however this sentence appears to place a considerable amount of blame on them.

- Overall, the results and discussion section should be thoroughly reviewed for sentence structure, clarity and flow.

Reviewer #3: PONE-D-24-04813R1

Thank for the opportunity to review the manuscript entitled “Examining the Relationship Between Language Development, Executive Function, and Screen Time: A Systematic Review” submitted to PLOS ONE.

This manuscript describes findings of a systematic review on the relation between Language Development, Executive Function, and Screen Time in children under six. These findings underscore the nuanced impact of screen time on language development and executive function, emphasizing the potential benefits of interactive and educational content when coupled with active parental engagement, while highlighting the risks associated with excessive passive screen time, particularly in lower SES contexts. While the paper addresses an important and relevant topic that is likely to interest many PLOS ONE readers, several issues need to be addressed, which are listed below:

1. I wanted to bring the authors’ attention to two recent meta-analyses on the very similar topic, which may enrich the discussion of the findings.

- Jing, M., Ye, T., Kirkorian, H. L., & Mares, M. L. (2023). Screen media exposure and young children's vocabulary learning and development: A meta‐analysis. Child Development, 94(5), 1398-1418.

- McHarg, G., Ribner, A. D., Devine, R. T., & Hughes, C. (2020). Screen time and executive function in toddlerhood: A longitudinal study. Frontiers in Psychology, 11, 570392.

2. The relation between the two research questions and the four key themes doesn’t appear straightforward to me. Is each research question examined under every theme? If so, why attention issue is an theme while executive function, which involves attention issues, is also part of the research question?

3. It is great that information about the studies’ methodology was included. However, from Table 2, it is unclear, at least for some studies, what exact research design is used. For example, are the first two studies (e.g., Zhang et al. and Hutton et al.) cross-sectional or longitudinal? This is critical information to be taken into account, because 1) previous meta-analyses show reliable difference between the effects of screen media exposure that are derived from different study designs (Jing, Ye, Kirkorian & Mares, 2023), and 2) it influences the coding of child age (e.g., in longitudinal research, the age when children are exposed to screen might be different from the age when the children are assessed on those outcome measures).

4. I would suggest the authors to add definitions for key concepts in the four themes. For example, what is exactly educational and what exactly interactive refers to? The meaning could vary by media format; educational TV is quite different from educational apps and games. And interactive media, which could either focus on physical interactivity or social contingency, is also too general to foster a meaningful discussion.

5. For the Methods, I have some qualm about the search strategy. Both language development and screen time are very board keywords, and I’m afraid this has missed eligible studies that use more specific narrow keywords, such as word learning, vocabulary learning or development, phonetic learning and TV viewing, video game playing, etc.

7. PLOS authors have the option to publish the peer review history of their article (what does this mean?). If published, this will include your full peer review and any attached files.

Reviewer #2: No

Reviewer #3: No

---

## [Author Response · Author response to Decision Letter 1]

1 Sep 2024

We would like to express our sincere gratitude to the reviewers for their continued valuable feedback and constructive criticism on our manuscript submitted to PLOS ONE. We have carefully considered each comment and recommendation provided during this second round of review and have made further revisions to our manuscript in response.

To ensure clarity and transparency in how we have addressed the reviewers' suggestions, we have prepared a detailed table that outlines each comment along with the corresponding revisions made in the manuscript.

We appreciate the opportunity to improve our manuscript, and we believe these revisions have further strengthened our work. We look forward to your further evaluation and hope our manuscript meets the journal's standards for publication.

Reviewer 2: “The study categorizes the impact into four key themes…” – it is not clear what ‘impact’ is referring to. I’m assuming this refers to the impact of screen time on language development and executive functioning, but this should be specified.

Authors’ Response: Based on your suggestion, we have revised the sentence to clearly state that the findings of the current study were categorized under four themes. The sentence now reads: "The findings of the current study were categorized under four themes." We believe this change addresses your concern and makes the intended meaning clearer (page 3).

Reviewer 2: The sentence ‘The prevalence of attention problems is found to be higher in individuals who engage in high levels of screen time’ appears out of place and could be removed from the abstract or included earlier on before discussing socioeconomic and cultural factors.

Authors’ Response: We have removed this sentence from its previous position and integrated the information earlier in the abstract, prior to the discussion of socioeconomic and cultural factors (page 3).

Reviewer 2: Overall, the introduction has been significantly improved and the authors should be commended on their excellent summary of the literature. The summary makes a clear case for RQ2 however, the authors have not really identified why RQ1 is needed as a separate aim. Is there previous evidence to say that quality of reporting in this field is poor and so that’s why we need it to be reviewed, or should this be a sub-aim that can be commented on within findings from RQ2? I would recommend listing RQ2 as the primary question and provide greater rationale on why RQ1 is needed.

Authors’ Response: In response to your suggestion, we have restructured our research questions to emphasize the primary focus of our study. As a result, we have integrated the original RQ1 ("What is the quality of reporting of studies on the relationship between screen time, language development, and executive function?") into the materials and methods section. We believe his revision allows us to maintain the focus on the primary research question while still acknowledging the significance of reporting quality as a critical aspect of our review. We hope this addresses your concerns and provides a clearer rationale for the structure of our study (pages 19-22).

Reviewer 2: 

1. Some minor edits to this section also include:

The summary of the link between language development and executive function on page 5 could be shortened to streamline this section.

2. Pg 8: “Mantilla and Edwards [51] noted that digital technology is now become and accepted an integral part…” should read “Mantilla and Edwards [51] noted that digital technology has now become and is accepted as an integral part…”

3. Noting the authors advised the incorrect placement of hyphens seems to be an issue due to the submission system, however I note one is still present on page 9, end of first paragraph ‘formulated’.

Authors’ Response:

Reviewer 2: 

Authors’ Response:

1. We have revised whole introduction section, and shortened our literature based on the feedback (pages 4-6). 

2. We have revised the sentence as suggested, so it now reads: "Mantilla and Edwards [51] noted that digital technology has now become and is accepted as an integral part..." (page 8).

3. We have corrected the hyphen in the word "formulated" as noted on page 9, at the end of the first paragraph.

Reviewer 2: The inclusion of Table 2 is very helpful and provides a good overview of the included studies. It does say in text that ‘author and year’ is described, however, currently the table only has the authors listed. Including details of the year of publication would be important especially given there was no restriction on publication date.

Authors’ Response: Thank you for your observation. We have updated Table 2 to include the year of publication along with the authors' names, as suggested (pages 12-17).

Reviewer 2: Please check the formatting of Table 3: (1) some of the study details in the first row are cut off; (2) the columns have not always been merged correctly for rows where headings are listed; (3) spelling error for 12a heading ‘Practisal’; (4) Rater agreement (%) should have the percentage sign after the number eg., 90%.

Authors’ Response:

1. Adjusted the first row so that all study details are fully visible (page 20).

2. Corrected the merging of columns where headings are listed (pages 20-21).

3. Fixed the spelling error for the "12a" heading from "Practisal" to "Practical." (page 21).

4. Updated "Rater agreement" to include the percentage sign after the numbers (e.g., 90%) (page 21).

Reviewer 2: Second sentence in the first paragraph on pg 22, should be reframed to state “Based on these data, all the articles have a value equal to or greater than 15 points.

Authors’ Response: We have revised the second sentence in the first paragraph on page 22 based on the reviewer's suggestion. The sentence now reads: “Based on these data, all the articles have a value equal to or greater than 15 points.”

Reviewer 2: 

1. Can the authors please clarify how screen time was incorporated into the Dolgikh et al [62] study? 

2. Did the extra classes that were attended in the study use educational screen time? The interpretation of this study may also need clarification, as the authors state ‘These findings suggest that structured extra classes can mitigate some of the negative effects of high screen time, enhancing language development and executive functions in preschool children’. However, it is not clear if that was actually tested in this research and is contradictory to the authors earlier statement that attending extra classes had no relationship to executive functioning?

Authors’ Response:

1. We have clarified this in the manuscript as below (page 24): 

In this study, screen time was assessed through a questionnaire administered to the mothers of the participating children. The questionnaire specifically asked about the average amount of screen time the children engaged in per week. The study compared screen time between two groups: children attending extra classes and those who did not. Although the difference in screen time between the groups was not statistically significant, it was noted that children in the extra classes group had slightly lower screen time (approximately 1016 minutes per week) compared to the no extra classes group (approximately 1299 minutes per week). 

This screen time data was considered in the analysis of the children’s executive functions, ensuring that its potential impact on cognitive development was accounted for in the study’s findings. (page 23).

2. We added further explanation regarding this feedback as below: 

“In this study, no specific data was collected regarding the use of screen time for educational purposes during the extra classes. In other words, the relationship between extra classes and screen usa was not directly examined. The study concluded that participation in extra classes had a positive impact on the development of children’s executive functions. Therefore, the study does not provide definitive conclusions on whether the supplementary classes balance out the negative effects of high screen time. On the other hand, the article did examine screen time as a factor related to the development of children’s executive functions, emphasizing that excessive screen time could have negative effects on the development of executive functions (pages 24-25).

Reviewer 2: “Parents adherence to screen time recommendations also enables beneficial characteristics of screen media, it was linked to better working memory” – what do the authors mean by ‘beneficial characteristics?

Authors’ Response: We have revised the manuscript to more clearly reflect this explanation and to ensure that the term "beneficial characteristics" accurately conveys the intended meaning on page 25.

Reviewer 2: I would recommend the authors rethink the wording of the theme ‘interactive and educational screen content’ to incorporate something about adherence to guidelines as some of the studies (e.g., Hutton et al) does not appear to investigate interactive or educational content specifically. Similarly, I would recommend reframing the theme ‘Parent and child interaction’ to ‘Parent-child interaction and the family context’ given much of the research summarised is focusing on socioeconomic status of the family.

Authors’ Response: Thank you for your valuable suggestions. We have revised the theme “Interactive and Educational Screen Content” to “Screen Content and Adherence to Guidelines” better reflect adherence to guidelines (page 23), and we have also rephrased the theme “Parent and Child Interaction” to “Parent-child Interaction and the Family Context” to align with the research focus on socioeconomic status (page 27).

Reviewer 2: First paragraph of pg 25 should be included in the Discussion section of the paper, rather than the results and it should also include references to some of the claims made (e.g., interactions can support children’s social and emotional development).

Authors’ Response: Following the reviewer’s suggestion, this section has been removed from the results section. The reviewer’s recommendation highlighted that this content was redundant. A similar discussion, already supported by relevant literature, is presented in the discussion section (pages 37-38).

Reviewer 2: As described on pg 29, the study by Kim and Chung [66] appear to include children outside the age range of the eligibility criteria (87.9months). Similarly for the study by Ribner et al [69] is outside the age range (7 years).

Authors’ Response: Although the study extends up to 87.9 months, we included the entire dataset in our analysis because it provides valuable insights into the longitudinal effects of screen time on child development. We added the following note to Table 2 (page 16): The progression from early childhood (starting at 5.5 months) to later stages (up to 87.9 months) allows for a comprehensive understanding of how screen exposure impacts cognitive and language development over time. Accordingly, we have revised the sentence on page 31 in the manuscript to clarify that Kim and Chung [66] conducted annual assessments from birth to 87.9 months, capturing the comprehensive longitudinal impact of screen time on child development as well.

Reviewer 2: Several statements in the second half of pg31 require associated references to support these claims.

Authors’ Response: The statements on page 33 are located in the results section and are intended to summarize the findings of the studies included in the theme. These findings were discussed based on the data from the studies themselves, which is why no additional literature references were included in this part of the text. To clarify the relationship between the results and the overarching theme, we have added transition phrases that link these findings to the data presented. Any discussion of these results in the context of existing literature has been deliberately reserved for the discussion section, where appropriate references have been included.

Reviewer 2: Final paragraph of the results section (pg 31-32) belongs within the discussion section.

Authors’ Response: In response, we have removed the final paragraph from the results section and integrated it into the appropriate theme within the discussion section (page 39).

Reviewer 2: The statement on pg 34 “… the results regarding the quality of reporting can inform researchers about the effective methods for data collection and analysis.” I don’t believe is correct. 

The review of adherence to reporting guidelines makes no comment on the effectiveness of methodology or data analysis. I would further argue that RQ1 does not make a substantial contribution to the existing body of literature and as a reader it is not clear what can be taken from the findings, other than research in this space seems to be generally applying the CONSORT checklist.

Authors’ Response: In response to your suggestion, we have restructured our research questions to better align with the primary focus of our study as we stated earlier. 

Consequently, we have integrated the original RQ1 (What is the quality of reporting of studies on the relationship between screen time, language development, and executive function?) into the materials and methods section (pages 19-21). 

This revision allowed us to maintain our focus on the primary research question and eliminate the discussed statement from the discussion section.

Reviewer 2: It would be helpful if the authors could make some comment on why there might be diversity in findings across studies within specific themes. For example, they describe that some studies find negative relationships and others found no relationship – but it is unclear why this may be the case. Are there differences in how they defined screentime or the age of the children they assessed?

Authors’ Response: We agree that explaining the diversity in findings across studies is important. In response, we added a discussion in the manuscript that considers variations in screen time definitions, the age of children, and the context and duration of screen exposure, which may account for the differing results (page 36).

Reviewer 2: I would recommend reframing the sentence “The literature also emphasizes that one of the significant variables affecting the negative impact of screen exposure is parents.” There is considerable context and factors outside of many parents control that may or may not influence their ability to support healthy screen time, however this sentence appears to place a considerable amount of blame on them.

Authors’ Response: We have revised the sentence to read: "The literature highlights the important role of parents in mitigating the negative effects of screen exposure, though this role can be influenced by various external factors (page 37).

Reviewer 2: Overall, the results and discussion section should be thoroughly reviewed for sentence structure, clarity and flow.

Authors’ Response: Discussion section has been revised (pages 33-40).

Reviewer 3: I wanted to bring the authors’ attention to two recent meta-analyses on the very similar topic, which may enrich the discussion of the findings.

- Jing, M., Ye, T., Kirkorian, H. L., & Mares, M. L. (2023). Screen media exposure and young children's vocabulary learning and development: A meta‐analysis. Child Development, 94(5), 1398-1418.

- McHarg, G., Ribner, A. D., Devine, R. T., & Hughes, C. (2020). Screen time and executive function in toddlerhood: A longitudinal study. Frontiers in Psychology, 11, 570392.

Authors’ Response: Thank you for bringing these recent meta-analyses to our attention. We have reviewed the studies by Jing et al. (2023) and McHarg et al. (2020) and have incorporated relevant insights from these works into the discussion of our findings (page 38).

Reviewer 3: The relation between the two research questions and the four key themes doesn’t appear straightforward to me. Is each research question examined under every theme? If so, why attention issue is an theme while executive function, which involves attention issues, is also part of the research question?

Authors’ Response: We have clarified the relationship between the research questions and the identified themes in our manuscript. Specifically, we have added an explanation to the discussion section that outlines why "Attent

---

## [Decision Letter · Decision Letter 2]

16 Oct 2024

PONE-D-24-04813R2Examining the Relationship Between Language Development, Executive Function, and Screen Time: A Systematic ReviewPLOS ONE

Dear Dr. Kara Aydemir,

Thank you for submitting your manuscript to PLOS ONE. After careful consideration, we feel that it has merit but does not fully meet PLOS ONE’s publication criteria as it currently stands. Therefore, we invite you to submit a revised version of the manuscript that addresses the points raised during the review process. Since this is another revision round in which some issues that have already been addressed by the reviewers and me, please pay close attention to all comments and try to answer them as unambigous as possible. I have noticed that you did not respond to my specific comments during your last revision. Therefore, I include them again here as well as below:

In line with the present reviews I recommend to especially pay close attention to the following points (as well as the reviewers detailed comments):

- Reference and discussion of other recent SR/MA in the field to highlight the relevance of your specific SR.

- While you elaborate in detail on various aspects surounding language development like executive functions, working memoriy, problem solving, processing etc. the link between these and "screen time" needs to be pointed out more clearly. At the moment, everything is listed rather additive but not put together into a strong line of argumentation and research rationale. Also, please try to avoid redudant argumentation in the regard, for instance, concerning the relation between language development, EF, and WM. What exactly is the important point here for your review?

- The overall aim of the SR is well described. Yet, the research questions do not seem to fully address what is (a) plannend to do and is (b) actually done by the SR. RQ1 could be omitted, since it appears to aim for a assesment of potential biases, which should be added to every SR following the PRISMA guidelines. RQ2 should be formulated as closely as possible to cover the scope of the SR, e.g.: "What is the relationship between children's screen time, language development, and executive functions reported in the present research literature?"

I hope these comments along with the reviewers' comments help to further improve your manuscript.

We look forward to receiving your revised manuscript.

Kind regards,

Alexander Röhm

Guest Editor

PLOS ONE

Journal Requirements:

**Additional Editor Comments:**

Dear Authors,

thank you again for your thourough revision of your manuscript. The remaining reviewer very much appreciates the efforts you put into your review and acknowledge the improvements it made since last revision. However, there are still some minor, but also major issues that deserve your full attention. In line with the present reviews I recommend to especially pay close attention to the following points (as well as the reviewers detailed comments):

- Reference and discussion of other recent SR/MA in the field to highlight the relevance of your specific SR.

- While you elaborate in detail on various aspects surounding language development like executive functions, working memoriy, problem solving, processing etc. the link between these and "screen time" needs to be pointed out more clearly. At the moment, everything is listed rather additive but not put together into a strong line of argumentation and research rationale. Also, please try to avoid redudant argumentation in the regard, for instance, concerning the relation between language development, EF, and WM. What exactly is the important point here for your review?

- The overall aim of the SR is well described. Yet, the research questions do not seem to fully address what is (a) plannend to do and is (b) actually done by the SR. RQ1 could be omitted, since it appears to aim for a assesment of potential biases, which should be added to every SR following the PRISMA guidelines. RQ2 should be formulated as closely as possible to cover the scope of the SR, e.g.: "What is the relationship between children's screen time, language development, and executive functions reported in the present research literature?"

I hope these comments along with the reviewers' comments help to further improve your manuscript.

Reviewers' comments:

Reviewer's Responses to Questions

**Comments to the Author**

1. If the authors have adequately addressed your comments raised in a previous round of review and you feel that this manuscript is now acceptable for publication, you may indicate that here to bypass the “Comments to the Author” section, enter your conflict of interest statement in the “Confidential to Editor” section, and submit your "Accept" recommendation.

Reviewer #3: (No Response)

2. Is the manuscript technically sound, and do the data support the conclusions?

Reviewer #3: Partly

3. Has the statistical analysis been performed appropriately and rigorously? 

Reviewer #3: N/A

4. Have the authors made all data underlying the findings in their manuscript fully available?

Reviewer #3: Yes

5. Is the manuscript presented in an intelligible fashion and written in standard English?

Reviewer #3: Yes

6. Review Comments to the Author

Reviewer #3: PONE-D-24-04813R2

I believe that the authors have made great effort improving the manuscript and addressing my major concerns. The flow of the introduction is stronger now, the method is more solid, and the discussion gets richer. However, there are still some remaining issues which are listed below:

I think the authors need to make an even stronger case for studying both language development and executive function in the context of screen time. The Introduction reviews the meaningful relationships between them, but it is not clear to me how this is reflected in the findings and discussion, which appears to address screen time’s relation with language development and executive function respectively.

Additionally, in addressing my previous concern about using broad keywords for study search, the authors stated that using very specific keywords could potentially result in inclusion of irrelevant studies, making the filtering process more problematic. I don't necessarily see that’s problematic, despite the fact that it will surely needs more work to screen relevant studies. At least, the method section needs to give clear definition and operational criteria for language development so that readers know what you mean by language development and how language development is examined in selected studies. And the same for screen time which is studied and measured in all different ways. I wound suggest to consider expanding Table 1 include that information and, if applicable, also for executive function as well.

Lastly, I think it would be beneficial for the study to be more mindful and clearer about the age range it focuses on when defining the research question, reviewing relevant literature in the introduction, and discussing and situating the findings. Given the remarkable developmental variability in children interacting with screen media, specifying the developmental stage could make the paper more informative and better connected to the larger literature.

7. PLOS authors have the option to publish the peer review history of their article (what does this mean?). If published, this will include your full peer review and any attached files.

Reviewer #3: No

---

## [Author Response · Author response to Decision Letter 2]

21 Oct 2024

Dear Dr. Alexander Röhm,

I am writing to submit the revised version of our manuscript titled “Examining the Relationship Between Language Development, Executive Function, and Screen Time: A Systematic Review” by Mazhar Bal, Ayşe Gül Kara Aydemir, Gülüzar Şule Tepetaş Cengiz, and Ahmet Altındağ for reconsideration for publication in PLOS ONE. We have carefully considered the editor’s and reviewer’s comments and suggestions, and we believe the revisions have significantly strengthened our manuscript. We deeply appreciate the time and effort dedicated by the reviewer and editorial board in evaluating our manuscript.

In response to both editorial and reviewer feedback, we have implemented comprehensive revisions throughout the manuscript. Following the editor's recommendations, we thoroughly restructured the introduction section to establish more explicit connections between screen time and language development, while incorporating a new section reviewing contemporary systematic reviews and meta-analyses to contextualize our study's contribution. We refined our research question to specifically address the relationship between screen time, language development, and executive functions in children aged 0-78 months. Addressing the reviewer's concerns, we enhanced Table 1 with detailed operational definitions and measurement criteria, systematically integrated age-range specifications throughout the manuscript, and strengthened the coherence between theoretical frameworks and empirical findings through enhanced analytical commentary in both the findings and discussion sections. These revisions collectively strengthen the manuscript's theoretical foundation, methodological clarity, and analytical depth while ensuring consistent developmental context throughout.

We believe that the revised manuscript now better meets the standards of PLOS ONE and addresses editor’s and reviewer’s concerns comprehensively. I appreciate your time and reconsideration and look forward to the opportunity of having our work reconsidered for publication in PLOS ONE. Ayşe Gül Kara Aydemir is the corresponding author of this study. Should you have any questions or require further information, please contact me via email at agkara@gmail.com

---

## [Editor Report · Decision Letter 3]

13 Nov 2024

Examining the Relationship Between Language Development, Executive Function, and Screen Time: A Systematic Review

PONE-D-24-04813R3

Dear Dr. Kara Aydemir,

We’re pleased to inform you that your manuscript has been judged scientifically suitable for publication and will be formally accepted for publication once it meets all outstanding technical requirements.

Kind regards,

Alexander Röhm

Guest Editor

PLOS ONE

Additional Editor Comments (optional):

Dear Authors,

thank you for this final revision of your manuscript, which now is strongly improved concerning the line of argumentation, scientific rigor, and presentation of the findings. I also very much appreciate the effort and work you put into this paper. There are only some last suggestions that you should consider to change before finalizing the manuscript:

1. I believe in Table 1 (p. 10) an AND/OR operand is missing between "Executive Function" "Children".

2. Would it be possible also to estimate the degree of interrater agreement for the quality codings (pp. 18ff) with a more than just percentages? I think, Krippendorff's Alpha should be the methodological standard here, which could be easily calculated with RECAL (https://dfreelon.org/utils/recalfront/).

3. I would suggest to conclude the article either with the Conclusion or Implications section, but not with the Limitations.

4. For full and transparent open science practice, please also add your search strategie, search terms/combination, and criteria to the OSF project.

5. Please, also consinder to publish your adapted CONSORT checklist.

I am very much looking forward to the published article, which, in my opinion, has considerable value for the further discussion of the important topic.

Best regards,

Alexander Röhm
---

## [Editor Report · Acceptance letter]

25 Nov 2024

PONE-D-24-04813R3 

PLOS ONE

Dear Dr. Kara Aydemir, 

I'm pleased to inform you that your manuscript has been deemed suitable for publication in PLOS ONE. Congratulations! Your manuscript is now being handed over to our production team.

Kind regards, 

on behalf of

Dr. Alexander Röhm 

Guest Editor

PLOS ONE